# An Optimization Model for Appraising Intrusion-Detection Systems for Network Security Communications: Applications, Challenges, and Solutions

**DOI:** 10.3390/s22114123

**Published:** 2022-05-29

**Authors:** Mohamed Abdel-Basset, Abduallah Gamal, Karam M. Sallam, Ibrahim Elgendi, Kumudu Munasinghe, Abbas Jamalipour

**Affiliations:** 1Faculty of Computers and Informatics, Zagazig University, Zagazig 44519, Egypt; mohamedbasset@ieee.org (M.A.-B.); abduallahgamal@fci.zu.edu.eg (A.G.); 2School of IT and Systems, University of Canberra, Canberra, ACT 2601, Australia; ibrahim.elgendi@canberra.edu.au (I.E.); kumudu.munasinghe@canberra.edu.au (K.M.); 3School of Electrical and Information Engineering, The University of Sydney, Sydney, NSW 2006, Australia; abbas.jamalipour@sydney.edu.au

**Keywords:** cyber-attacks, intrusion-detection system, MCDM, q-rung orthopair fuzzy sets, q-ROFWG

## Abstract

Cyber-attacks are getting increasingly complex, and as a result, the functional concerns of intrusion-detection systems (IDSs) are becoming increasingly difficult to resolve. The credibility of security services, such as privacy preservation, authenticity, and accessibility, may be jeopardized if breaches are not detected. Different organizations currently utilize a variety of tactics, strategies, and technology to protect the systems’ credibility in order to combat these dangers. Safeguarding approaches include establishing rules and procedures, developing user awareness, deploying firewall and verification systems, regulating system access, and forming computer-issue management groups. The effectiveness of intrusion-detection systems is not sufficiently recognized. IDS is used in businesses to examine possibly harmful tendencies occurring in technological environments. Determining an effective IDS is a complex task for organizations that require consideration of many key criteria and their sub-aspects. To deal with these multiple and interrelated criteria and their sub-aspects, a multi-criteria decision-making (MCMD) approach was applied. These criteria and their sub-aspects can also include some ambiguity and uncertainty, and thus they were treated using q-rung orthopair fuzzy sets (q-ROFS) and q-rung orthopair fuzzy numbers (q-ROFNs). Additionally, the problem of combining expert and specialist opinions was dealt with using the q-rung orthopair fuzzy weighted geometric (q-ROFWG). Initially, the entropy method was applied to assess the priorities of the key criteria and their sub-aspects. Then, the combined compromised solution (CoCoSo) method was applied to evaluate six IDSs according to their effectiveness and reliability. Afterward, comparative and sensitivity analyses were performed to confirm the stability, reliability, and performance of the proposed approach. The findings indicate that most of the IDSs appear to be systems with high potential. According to the results, Suricata is the best IDS that relies on multi-threading performance.

## 1. Introduction

The continuous development of computer systems has led to the increasing dependence of companies, organizations, and people on computer networks in performing their functions and offering their services in modern ways [1]. However, at the same time, it has become vulnerable to penetration by attackers with the aim of making illegal gains by exploiting some security vulnerabilities, which led to an increase in interest in issues of protection and security of these systems. Today there are many methods used within this field, and there are many intrusion-detection systems (IDSs) available [2]. IDSs are a necessity for the stability of an organization’s normal system performance. In this regard, traditional intrusion-detection techniques are highly unrewarding and ineffective due to the multiplicity of attack methods and their different forms. Among the traditional methods used previously are obfuscation, transformation, and polymorphism techniques, which lead to malware resistance [3]. Despite the prominent role it plays, it still has some shortcomings. Therefore, there was a need to continue conducting research on intrusion-detection systems in order to reach an optimal structure that achieves a high protection rate.

The internet changed the concept of computing as we know it. The possibilities and opportunities available became unlimited, and with it, the risks and opportunities for breakthroughs increased. Computer security primarily focuses on protecting a specific source or valuable data and information within a single computer device. Security is defined as the reaction taken to security threats resulting from a harmful act by some people. The value of the data can be violated in three ways: privacy, integrity, and availability of information [4]. Computer protection is generally referred to by the term CIA, which is represented by the following three concepts: Confidentiality: Preventing unauthorized persons from disclosing or accessing information; i.e., accessing information only by authorized persons.Integrity: Maintaining information integrity by preventing unauthorized modification.Availability: It is the ability of a computer to work and provide the resources and services expected of it to legitimate people upon request.

Network security includes all actions or activities taken by organizations and companies in order to protect resources and ensure the integrity and continuity of operations across networks [5]. Security policies also define the permissions available to users in the way they use network components and resources. In order to build an effective network-protection strategy, all potential security threats must be identified, and then the most effective set of tools to combat them must be selected [6]. Preventing all exploits for vulnerabilities in networks and systems is not possible [7]. Network protection is achieved through the use of a set of components at several levels, with the aim of protecting organizations from internal attacks and external attacks as much as possible. A firewall is a component that achieves the most basic level of protection but is not sufficient on its own. Designing and implementing a completely secure system is very difficult in practice, but it is possible to detect intrusions and take appropriate measures to protect against them. This is what the IDS basically does, as it is used as an alert system, within the security and protection system, that gives an alert when it detects an attempt by someone to penetrate the computer system or network [8]. As a result, IDSs are important in a network security solution. The primary goal of IDSs is to detect an intrusion while it is occurring rather than after it has ended, and then alert the person responsible for the problem by sending an email or setting off an alert. It must be able to take any action to minimize harm to the system due to the hack. The second goal is to collect data from the system, record all important events, and determine the source of the attack, and these data are used for legal purposes as evidence or proof against the attacker.

IDSs must be placed in strategic places so that they are able to see network traffic in order to analyze it and thus achieve the maximum benefit from it [9]. In this regard, several ways to classify IDSs are described using different analysis and control methods. The most common way to classify intrusion-detection systems is to group them according to the location of the information source. Basic information sources are network packets captured from a network backbone or local network segments, operating systems, and critical files. Intrusion-detection systems can be classified into host-based detection-intrusion systems, network-based intrusion-detection systems, and hybrid systems. On the user’s computer, a host-based intrusion-detection system (HIDS) is installed. HIDSs operate on information collected from within a single computer system [10]. HIDSs employ monitoring sensors, also called clients, on each host to be monitored. In general, the most common forms of information sources for HIDSs are operating system audit logs, system logs, and critical system files [10]. The customer checks these sources for unauthorized changes or patterns of suspicious activity. This allows HIDSs to reliably analyze activities, accurately identifying which users and which processes are participating in a specific attack on the operating system. The most common form of IDSs is network-based intrusion-detection systems (NIDSs) [11]; also, most companies and organizations are often supported by NIDSs along with firewalls. These systems detect attacks by capturing and analyzing network packets by listening to network segments or switches [11]. In this case, the system is placed on an entire network segment and not on a single device within the network, or it is placed to monitor a gateway on the switch. Thus, it can monitor all mobile packets between groups of computers connected to the network, by matching one or more packets with the database of signatures of attacks, or by analyzing the traffic to detect anomalies. NIDSs can be taken advantage of by placing it outside of firewalls, thus alerting the responsible person to incoming packets that might circumvent the firewall. Both HIDSs and NIDSs have strengths and benefits that complement each other [12]. Figure 1 introduces the general architecture of an HIDS and NIDS. The next generation of IDSs must combine the two technologies in order to improve the network’s resistance to attacks and abuse. In addition, they should enhance security policy and provide greater flexibility in application and deployment options. A hybrid IDS is a mixture of an HIDS and NIDS. It provides a combination of the strengths of the two methods. Their modus operandi varies from product to product, making it difficult to define and determine hybrid intrusion systems in a more accurate manner.

Afterward, detection methods are the basis of intrusion-detection techniques, which are the engine in detecting the malicious activities of the information source. Detection methods analyze the information they monitor and trigger alerts if malicious traffic is detected. Accordingly, IDSs can be categorized, according to the detection methods used, into anomaly-based intrusion-detection systems (AIDSs) [13] and signature-based intrusion-detection systems (SIDSs) [14]. In this regard, AIDSs operate on the assumption that malicious events are different from normal actions, and thus the differences are sought to detect the attack. These systems constitute profiles of historical data collected during a period of normal operation. It then collects event data to determine when the monitored activity deviates from normal behavior and triggers an alarm. SIDSs also are called signature-based detection because alerts are generated based on the signatures of specific attacks. This type of signature attack involves specific traffic or activity based on known hacking activity. It is also called misuse-based detection. The basic premise is the model that has been written to describe bad behavior, after which the system compares the sequence of information with this model to decide what is normal and what is malicious. These systems are accurate and emit fewer false alarms, but they do not detect a hack unless there is a predetermined model for it.

Nowadays, Internet networks are vulnerable to a wide range of threats and attacks, such as impersonation, privilege breaches, data loss, altered and fraudulent data units, and denial of connections [15]. Therefore, IDSs have an essential task to protect the normal system performance of an organization. It is, therefore, necessary to add new security requirements and additional networking measures to the network security requirements [16]. IDSs must constantly change and adapt to all these new threats and assault technologies. Therefore, the process of determining the best IDS for network protection, threat warning, and cyber-attacks is a very difficult task in light of the various criteria on which an IDS is developed. Thus, IDSs should not be chosen to quickly secure the network without a thorough understanding of the technology, solutions, and potential consequences.

In this study, a set of criteria were adopted to evaluate IDSs according to previous studies and expert opinions. The criteria that have been adopted were divided into four basic criteria—protected system, audit source location, alerts, and types—and each main criterion includes several sub-aspects. The set of sub-aspects are as follows: HIDS, NIDS, hybrids, host log files, network packets, application log files, IDS sensor application, network, host, open-source, closed source, and freeware. In order to solve such complex problems related to the evaluation of IDSs, multi-criteria decision-making (MCDM) has been proven to be one of the best tools for the effective evaluation of IDS [16]. MCDM is popular in complex problems because it enables the decision-maker to take care of all the available criteria and take an appropriate decision as per the priority [17]. Since the ideal choice is governed by multiple criteria, a good decision-maker, in certain situations, may look for criteria of high impact on which to focus.

Consequently, due to the assessment of IDSs under multiple criteria and a pluralistic viewpoint, the assessment process is tainted by ambiguity and uncertainty, which is difficult to deal with in real numbers. Hence, the q-rung orthopair fuzzy sets (q-ROFSs) theory has been applied to deal with such complex problems [18]. The q-ROFS proved to be effective in solving ambiguous and uncertain problems as it came as a generalization of the intuitionistic fuzzy sets (IFSs) [19] and Pythagorean fuzzy sets (PFSs) theories [20].

Finally, to deal with the problem of evaluating the effectiveness of IDSs, a hybrid approach consisting of two multi-criteria decision-making methods, the entropy method [21] and the combined compromised solution (CoCoSo) method [22], was adopted. The proposed hybrid approach is presented under the q-rung orthopair fuzzy environment and by utilizing the q-rung orthopair fuzzy numbers (q-ROFNs) numbers. Firstly, the entropy method was adopted to evaluate the main and sub-aspects and to determine the final weights. Secondly, the CoCoSo method was applied to evaluate the available alternatives and determine the best alternative.

The main contributions of this study can be listed as follows: (i) a novel hybrid MCDM approach is proposed for the evaluation of the IDS; (ii) this hybrid MCDM approach is named q-ROF entropy-CoCoSo, which give assessments of subjective and impartial expert insights; (iii) the q-ROFSs method is conducted to handle the uncertainty in experts’ evaluations; (iv) a q-ROF entropy is utilized to compute the criteria weights; and (v) a q-ROF CoCoSo is suggested to evaluate the selected IDSs.

The article is organized as follows: Section 2 highlights the IDS and insights into MCDM; Section 3 introduces some preliminaries of q-ROFS and the proposed research approach; Section 4 illustrates the application of the MCDM approach in evaluating the IDS and discussion; and Section 5 contains the conclusions.

**Figure 1 sensors-22-04123-f001:**
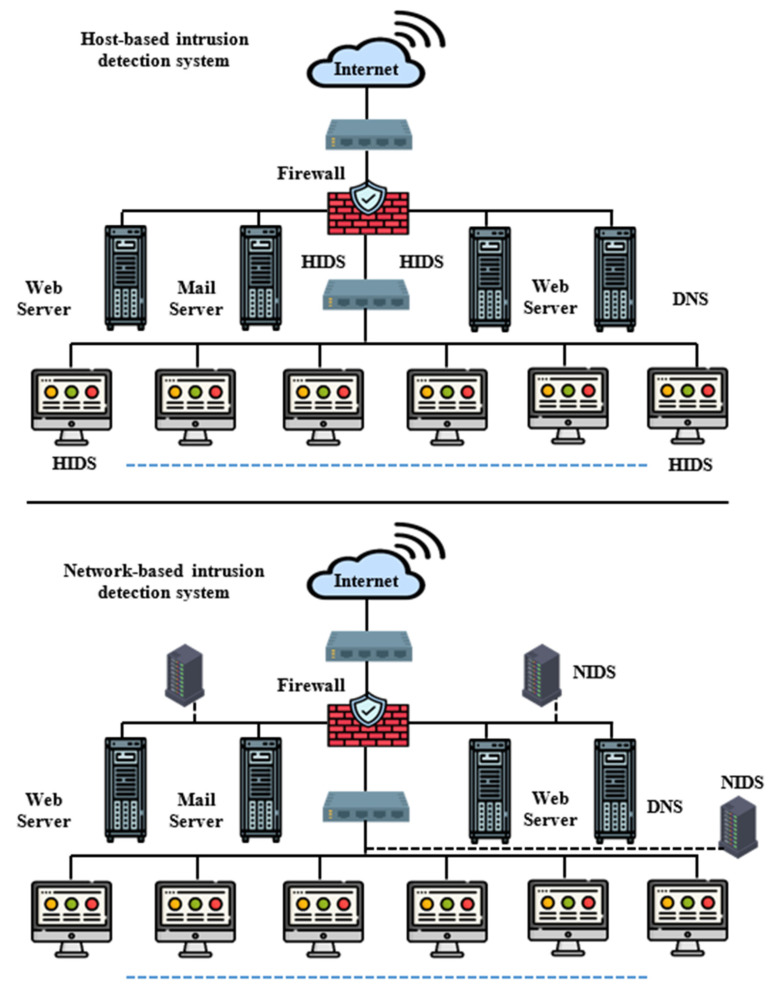
Comparison of the HIDS and NIDS structure [23].

## 2. Background Information

This section provides basic knowledge about IDSs to enable a deeper understanding of this topic; also, some basic information related to MCDM is presented. Afterward, some studies related to q-ROF theory are introduced. Alyami et al. presented a study based on a hybrid MCDM approach consisting of the fuzzy analytical hierarchy process (AHP) and the fuzzy technique for order performance by similarity to ideal solution (TOPSIS) to evaluate the effectiveness of the IDSs [16]. In their study, they used four main criteria and thirteen sub-aspects to evaluate five IDSs. Their results indicate that Suricata is the most effective IDS. Their results also indicate that most of the IDSs that were evaluated in the study are effective and close in their results. Abushark et al. developed a study to evaluate the optimization of machine learning-based IDSs using a hybrid MCDM approach that comprises the AHP and TOPSIS in a fuzzy environment [24]. Their findings aim to identify attributes related to cyber security, allowing the design of more effective and efficient IDSs. Al-Harbi et al. presented a study for an optimal evaluation of machine learning-based IDSs using a hybrid MCDM model that includes AHP and TOPSIS under hesitant fuzzy conditions [8]. Their findings aim to identify features related to cyber security, allowing the design of more effective and IDSs. Almotiri presented an evaluation system to detect malicious traffic based on system performance [25]. They adopted an MCDM approach consisting of AHP–TOPSIS methods to rank the impact of alternatives according to their overall performance. Their study aims to be a reference for practitioners working in the field of evaluating and selecting the most effective traffic detection approach.

Afterward, some studies related to MCDM approaches and their applications in different fields were presented. Sharma and Kaul presented a study to deal with network performance delays and disruptions due to cluster-based communications that place a significant burden on the cluster head (CH) [26]. They used an MCDM approach including two AHP–TOPSIS methods to reduce the overburden on a single CH through a multi-CH scheme. Ogundoyin and Kamil presented a study to address security and privacy issues where fog servers can be used to process private and respond to time-sensitive information [27]. They applied the fuzzy AHP MCDM approach to determine and prioritize confidence parameters in fog computing. Their results indicate that quality of service is the best priority parameter that a service requester can use to evaluate the trusted standard of a service provider. Kumar et al. introduced a study to evaluate the impact of various malware analysis methods on the perspective of web applications [28]. They applied an MCDM approach that comprises AHP and TOPSIS methods under a fuzzy environment. Their results indicate that reverse engineering is the most effective method for analyzing complex malware.

There are many theories dealing with uncertainty, including the q-ROFS theory. Thus, we present some related studies as follows. Duane et al. introduced a study to deal with risks in information sharing and software piracy, which poses a threat to any system [29]. They applied the q-rung orthopair double hierarchy linguistic term set (q-RODHLTS) in the MCDM process. To prove the validity of their results, they applied the proposed approach to many information security systems. Panetikul et al. presented a study to analyze computer security threat analysis and control under a q-ROF environment [30]. They applied an approach based on the combination of the Heronian mean (HM) operator with complex q-ROFS is to initiate the complex q-rung orthopair fuzzy HM (Cq-ROFHM) operator. To prove the reliability and efficiency of the techniques used, some illustrative examples are introduced. Cheng et al. presented a study for evaluating sustainable enterprise risk management in manufacturing small and medium-sized enterprises [31]. They adopted a new extended Vlse Kriterijumska Optimizacija Kompromisno Resenje (VIKOR) approach using q-ROFSs. Peng et al. introduced a study for presenting a new score function of q-ROFN for solving the failure issues when comparing two q-ROFNs [32].

Finally, given the importance of IDSs and the importance of implementing them in a manner appropriate to their specific situation, choosing the most effective and appropriate one is a great challenge. Hence, there is an urgent and great need to evaluate IDSs. In this regard, a set of main and sub-criteria affecting the selection of the most effective IDS is identified; also, a set of alternatives are identified to be evaluated according to these criteria, using an MCDM approach and under a q-ROF environment.

## 3. Proposed Research Approach

### 3.1. Prelimianries

In this section, we list some concepts, procedures, and fundamental definitions related to IFSs, PFSs, and q-ROFSs.

#### 3.1.1. Intuitionistic Fuzzy Sets

Atanassov developed IFSs as an extension of fuzzy set theory in 1986. IFSs are distinguished by the grade of membership and the grade of non-membership when their total is 1 or less than 1. It is explained as stated in Definition 1 [19].

**Definition** **1.**
*Let be Ҳ a fixed set. An IFS *

$\tilde{I}$

* in Ҳ is an entity having the form given by*

(1)
$$\tilde{I}=\left\{\left(ҳ,\;\mu_{\tilde{I}}\left(ҳ\right),\;{ʋ}_{\tilde{I}}\left(ҳ\right)\right)|\;ҳ∊Ҳ\right\}$$

*where the function*

$\mu_{\tilde{I}}$

*: Ҳ → [0, 1] describes the grade of membership of an element to the sets*

$\tilde{I}$

*and*

${ʋ}_{\tilde{I}}$

*: Ҳ → [0, 1] describes the grade of non-membership of an element to the sets*

$\tilde{I}$

*, with the condition that*


(2)
$$0\;\leq \;\mu_{\tilde{I}}\left(ҳ\right)+{ʋ}_{\tilde{I}}\left(ҳ\right)\;\leq \;1,\;\mathrm{for}\;Ɐ\;ҳ∊Ҳ$$


*The grade of hesitancy is computed as follows:*

(3)
$${Ԉ}_{\tilde{I}}\left(ҳ\right)=1-\mu_{\tilde{I}}\left(ҳ\right)-{ʋ}_{\tilde{I}}\left(ҳ\right)$$



**Definition** **2.**
*Let *

$\tilde{₳}$

* = (*

$\mu_{\tilde{₳}}$

*, *

${ʋ}_{\tilde{₳}}$

*) and *

$\tilde{Ƀ}$

* = (*

$\mu_{\tilde{Ƀ}}$

*, *

${ʋ}_{\tilde{Ƀ}}$

*) be two intuitionistic fuzzy numbers (IFNs), then the addition and multiplication operations on these two IFNs as follows:*

(4)
$$\tilde{₳}⊕\;\tilde{Ƀ=\left(\mu_{\tilde{₳}}+{\;\mu }_{\tilde{Ƀ}}-{\;\mu }_{\tilde{₳}}\mu_{\tilde{Ƀ}},\;{ʋ}_{\tilde{₳}}{ʋ}_{\tilde{Ƀ}}\right)}$$


(5)
$$\tilde{₳}⊗\tilde{Ƀ}=(\mu_{\tilde{₳}}\mu_{\tilde{Ƀ}},\;{ʋ}_{\tilde{₳}}+{ʋ}_{\tilde{Ƀ}}-{ʋ}_{\tilde{₳}}{ʋ}_{\tilde{Ƀ}})\;$$



#### 3.1.2. Pythagorean Fuzzy Sets

Yager developed PFSs as an extension of the IFSs [20]. They are distinguished by two membership grades termed as membership and non-membership. The total membership and non-membership grades in PFSs are unlike in IFSs. In PFSs, the membership and non-membership grade may be more than 1, but the total of their squares has to be at most 1. It is explained as stated in Definition 3.

**Definition** **3.**
*Let be Ҳ a fixed set. A PFS *

$\tilde{P}$

* in Ҳ is an entity having the form given by*

(6)
$$\tilde{Ҏ}=\left\{\left(ҳ,\;\mu_{\tilde{Ҏ}}\left(ҳ\right),\;{ʋ}_{\tilde{Ҏ}}\left(ҳ\right)\right)|ҳ∊Ҳ\right\}$$

*where the function *

$\mu_{\tilde{Ҏ}}$

*: Ҳ → [0, 1] describes the grade of membership of an element *

ҳ∊Ҳ

* to the sets *

$\tilde{Ҏ}$

* and *

${ʋ}_{\tilde{Ҏ}}$

*: Ҳ → [0, 1] describes the grade of non-membership of an element *

ҳ∊Ҳ

* to the sets *

$\tilde{Ҏ}$

*, with the condition that*

(7)
$$0\;\leq \;\left(\mu_{\tilde{Ҏ}}\left(ҳ\right)\right){}^{2}+\left({ʋ}_{\tilde{Ҏ}}\left(ҳ\right)\right){}^{2}\;\leq \;1,\;\mathrm{for}\;Ɐ\;ҳ∊Ҳ$$


*The grade of uncertainty is computed as follows:*

(8)
$${Ԉ}_{\tilde{Ҏ}}\left(ҳ\right)=\sqrt{1-{\;\mu }_{\tilde{Ҏ}}\left(ҳ\right){}^{2}-{ʋ}_{\tilde{Ҏ}}\left(ҳ\right){}^{2}}$$



**Definition** **4.**
*Let*

${\tilde{Ҏ}}_{1}$

* = (*

$\mu_{{\tilde{Ҏ}}_{1}}$

*, *

${ʋ}_{{\tilde{Ҏ}}_{1}}$

*) and *

${\tilde{Ҏ}}_{2}$

* = (*

$\mu_{{\tilde{Ҏ}}_{2}}$

*, *

${ʋ}_{{\tilde{Ҏ}}_{2}}$

*) be two *
*Pythagorean fuzzy numbers (PFNs), then the addition and multiplication operations on these two PFNs as follows:*

(9)
$${\tilde{Ҏ}}_{1}⊕{\tilde{Ҏ}}_{2}=\left(\sqrt{\mu_{{\tilde{Ҏ}}_{1}}{}^{2}+\mu_{{\tilde{Ҏ}}_{2}}{}^{2}-\mu_{{\tilde{Ҏ}}_{1}}{}^{2}\mu_{{\tilde{Ҏ}}_{2}}{}^{2}\;},{ʋ}_{{\tilde{Ҏ}}_{1}}{ʋ}_{{\tilde{Ҏ}}_{2}}\right)$$


(10)
$${\tilde{Ҏ}}_{1}⊗{\tilde{Ҏ}}_{2}=\left(\mu_{{\tilde{Ҏ}}_{1}}\mu_{{\tilde{Ҏ}}_{2}},\sqrt{{ʋ}_{{\tilde{Ҏ}}_{1}}{}^{2}+{ʋ}_{{\tilde{Ҏ}}_{2}}{}^{2}-{ʋ}_{{\tilde{Ҏ}}_{1}}{}^{2}{ʋ}_{{\tilde{Ҏ}}_{2}}{}^{2}\;}\right)$$



#### 3.1.3. Q-Rung Orthopair Fuzzy Sets

Yager presented q-ROFSs in 2018 with the grade of membership and non-membership. In q-ROFSs, the total of the *q*th power of the membership and non-membership grades should be at most equal to 1 [18]. In Figure 2, it is readily noted that q-ROFSs have a reasonable membership degree extent greater than that of the IFSs and PFSs. q-ROFSs are explained as stated in Definition 5.

**Definition** **5.**
*A q-ROFS*

$\overset{˘}{Ҩ}$

*in a finite universe of discourse*

Ҳ

*=*

$x_{1}$

*,*

$x_{2}$

*, …,*

$\;x_{n}$

*is defined by Yager as follows [18]:*

(11)
$$\overset{˘}{Ҩ}=\left\{\left(ҳ,\;\mu_{\overset{˘}{Ҩ}}\left(ҳ\right),{ʋ}_{\overset{˘}{Ҩ}}\left(ҳ\right)\right)|ҳ∊Ҳ\right\}$$

*where the function *

$\mu_{\tilde{Ҩ}}$

*: Ҳ → [0, 1] defines the grade of membership of an element *

ҳ∈Ҳ

* to the sets *

$\tilde{Ҩ}$

* and *

${ʋ}_{\tilde{Ҩ}}$

*: Ҳ → [0, 1] defines the grade of non-membership of an element *

ҳ∈Ҳ

* to the sets *

$\tilde{Ҩ}$

*, with the condition that*

(12)
$$0\;\leq \;\mu_{\tilde{Ҩ}}\left(ҳ\right){}^{q}+{ʋ}_{\tilde{Ҩ}}\left(ҳ\right){}^{q}\;\leq \;1,\;\mathrm{for}\;Ɐ\;ҳ\in Ҳ$$


*The grade of uncertainty is computed as follows*

(13)
$${Ԉ}_{\tilde{Ҩ}}\left(ҳ\right)=\sqrt[{q}]{1-{\;\mu }_{\tilde{Ҩ}}\left(ҳ\right){}^{q}-{ʋ}_{\tilde{Ҩ}}\left(ҳ\right){}^{q}}$$



**Definition** **6.**
*Let*

$\overset{˘}{Ҩ}$

*= (*

μҨ˜

*,*

${ʋ}_{\tilde{Ҩ}}$

*),*

${\overset{˘}{Ҩ}}_{1}$

*= (*

$\mu_{{\tilde{Ҩ}}_{1}}$

*,*

${ʋ}_{{\tilde{Ҩ}}_{1}}$

*),*

${\overset{˘}{Ҩ}}_{2}$

*= (*

$\mu_{{\tilde{Ҩ}}_{2}}$

*,*

${ʋ}_{{\tilde{Ҩ}}_{2}}$

*), be three q-ROFNs, then their procedures can be well-defined as follows [18]:*

(14)
$${\overset{˘}{Ҩ}}_{1}\;\cap \;{\overset{˘}{Ҩ}}_{2}=\left(\min\left\{\mu_{{\tilde{Ҩ}}_{1}},{\;\mu }_{{\tilde{Ҩ}}_{2}}\right\},\max\left\{{ʋ}_{{\tilde{Ҩ}}_{1}},\;{ʋ}_{{\tilde{Ҩ}}_{2}}\right\}\right)$$


(15)
$${\overset{˘}{Ҩ}}_{1}\cup^{\;}{\overset{˘}{Ҩ}}_{2}=\left(\max\left\{\mu_{{\tilde{Ҩ}}_{1}},{\;\mu }_{{\tilde{Ҩ}}_{2}}\right\},\min\left\{{ʋ}_{{\tilde{Ҩ}}_{1}},\;{ʋ}_{{\tilde{Ҩ}}_{2}}\right\}\right)\;$$


(16)
$${\overset{˘}{Ҩ}}_{1}⊕{\overset{˘}{Ҩ}}_{2}=\left({\left(\mu_{{\tilde{Ҩ}}_{1}}{}^{q}+\mu_{{\tilde{Ҩ}}_{2}}{}^{q}-\mu_{{\tilde{Ҩ}}_{1}}{}^{q}\mu_{{\tilde{Ҩ}}_{2}}{}^{q}\right)}^{\frac{1}{q}},\;{ʋ}_{{\tilde{Ҩ}}_{1}}{ʋ}_{{\tilde{Ҩ}}_{2}}\right)$$


(17)
$${\overset{˘}{Ҩ}}_{1}⊗{\overset{˘}{Ҩ}}_{2}=\left(\mu_{{\tilde{Ҩ}}_{1}}\mu_{{\tilde{Ҩ}}_{2}},\;{\left({ʋ}_{{\tilde{Ҩ}}_{1}}{}^{q}+{ʋ}_{{\tilde{Ҩ}}_{2}}{}^{q}-{ʋ}_{{\tilde{Ҩ}}_{1}}{}^{q}{ʋ}_{{\tilde{Ҩ}}_{2}}{}^{q}\right)}^{\frac{1}{q}}\right)$$


(18)
$$\lambda \overset{˘}{Ҩ}=\left({\left(1-{\left(1-\mu_{\overset{˘}{Ҩ}}{}^{q}\right)}^{\lambda }\right)}^{\frac{1}{q}},{ʋ}_{\overset{˘}{Ҩ}}{}^{\lambda }\right),\;\lambda >0$$


(19)
$${\overset{˘}{Ҩ}}^{\lambda }=\left(\mu_{\overset{˘}{Ҩ}}{}^{\lambda },{\left(1-{\left(1-{ʋ}_{\overset{˘}{Ҩ}}{}^{q}\right)}^{\lambda }\right)}^{\frac{1}{q}}\right),\;\lambda >0$$



**Definition** **7.**
*Let*

$\overset{˘}{Ҩ}$

*= (*

μҨ˜

*,*

${ʋ}_{\tilde{Ҩ}}$

*) be a q-ROFN; the score function S(*

$\overset{˘}{Ҩ}$

*) of*

$\overset{˘}{Ҩ}$

*can be expressed as in [33], and the accuracy function A(*

$\overset{˘}{Ҩ}$

*) of*

$\overset{˘}{Ҩ}$

*can be well defined, as in [34], shown by Equations (20) and (21), respectively.*

(20)
$$S(\overset{˘}{Ҩ})=\frac{1}{2}\left(1+\mu_{\overset{˘}{Ҩ}}{}^{q}-{ʋ}_{\overset{˘}{Ҩ}}{}^{q}\right)$$


(21)
$$A(\overset{˘}{Ҩ})=\mu_{\overset{˘}{Ҩ}}{}^{q}+{Ҩ}_{\overset{˘}{Ҩ}}{}^{q}$$



**Definition** **8.***Let*${\overset{˘}{Ҩ}}_{i}$*=*$\left(\mu_{{\tilde{Ҩ}}_{i}},\;{ʋ}_{{\tilde{Ҩ}}_{i}}\right)\;$*(i = 1, 2, … n) be set of q-ROFNs and Ⱳ =*${\left({ⱳ}_{1},{ⱳ}_{2},\;\ldots ,{ⱳ}_{n}\;\right)}^{T}$*be weight vector of*${\overset{˘}{Ҩ}}_{i}$*with*$\sum_{i=1}^{n}{Ⱳ}_{i}$*= 1 and*${Ⱳ}_{i}$ ∈*[0, 1]. Q-rung orthopair fuzzy weighted average (q-ROFWA) and q-rung orthopair fuzzy weighted geometric (q-ROFWG) operators can be expressed as in [34], shown by Equations (22) and (23), respectively.*(22)$$q\text{-}\;\mathrm{ROFWA}\left({\overset{˘}{Ҩ}}_{1},{\overset{˘}{Ҩ}}_{2},\ldots ,\;{\overset{˘}{Ҩ}}_{n}\right)=\left({\left(1-\prod_{i=1}^{n}{\left(1-\mu_{{\tilde{Ҩ}}_{i}}{}^{q}\right)}^{{Ⱳ}_{i}}\right)}^{\frac{1}{q}},\;\prod_{i=1}^{n}{ʋ}_{{\tilde{Ҩ}}_{i}}{}^{{ⱳ}_{i}}\right)$$

(23)
$$q\text{-}\;\mathrm{ROFWA}\left({\overset{˘}{Ҩ}}_{1},{\overset{˘}{Ҩ}}_{2},\ldots ,\;{\overset{˘}{Ҩ}}_{n}\right)=\left(\prod_{i=1}^{n}\mu_{{\tilde{Ҩ}}_{i}}{}^{{Ⱳ}_{i}},\;{\left(1-\prod_{i=1}^{n}{\left(1-{ʋ}_{{\tilde{Ҩ}}_{i}}{}^{q}\right)}^{{Ⱳ}_{i}}\right)}^{\frac{1}{q}}\right)$$



**Definition** **9.***Darko and Liang developed an operator named the weighted q-rung orthopair fuzzy Hamacher average (Wq-ROFHA)* [35] *as in Equations (24) and (25). Let*
${\overset{˘}{Ҩ}}_{i}$
*=*
$\left(\mu_{{\tilde{Ҩ}}_{i}},\;{ʋ}_{{\tilde{Ҩ}}_{i}}\right)$* (i = 1, 2, … n) be set of q-ROFNs and Ⱳ =*
${\left({ⱳ}_{1},{ⱳ}_{2},\;\ldots ,{ⱳ}_{n}\;\right)}^{T}$
*be a weight vector of*
${\overset{˘}{Ҩ}}_{i}$
*with*
$\sum_{i=1}^{n}{Ⱳ}_{i}$
*= 1 and*
${Ⱳ}_{i}$ ∈
*[0, 1].*
(24)$$\mathrm{Wq}\text{-}\;\mathrm{ROFHA}\left({\overset{˘}{Ҩ}}_{1},{\overset{˘}{Ҩ}}_{2},\ldots ,\;{\overset{˘}{Ҩ}}_{n}\right)={ⱳ}_{1}\left({\overset{˘}{Ҩ}}_{1}\right)⊕{ⱳ}_{2}\left({\overset{˘}{Ҩ}}_{2}\right)⊕\ldots \;⊕{ⱳ}_{n}\left({\overset{˘}{Ҩ}}_{n}\right)=\begin{array}{c} n\\ ⊕\\ i=1 \end{array}{ⱳ}_{i}\left({\overset{˘}{Ҩ}}_{i}\right)$$
(25)$$\begin{array}{c} \mathrm{Wq}\text{-}\;\mathrm{ROFHA}\left({\overset{˘}{Ҩ}}_{1},{\overset{˘}{Ҩ}}_{2},\ldots ,\;{\overset{˘}{Ҩ}}_{n}\right)=\\ (\sqrt[{q}]{\frac{\prod_{i=1}^{n}{\left(1+\left(\gamma -1\right){\left(\mu_{{\overset{˘}{Ҩ}}_{i}}\right)}^{q}\right)}^{{Ⱳ}_{i}}-\prod_{i=1}^{n}{\left(1-{\left(\mu_{{\overset{˘}{Ҩ}}_{i}}\right)}^{q}\right)}^{{ⱳ}_{i}}}{\prod_{i=1}^{n}{\left(1+\left(\gamma -1\right){\left(\mu_{{\overset{˘}{Ҩ}}_{i}}\right)}^{q}\right)}^{{Ⱳ}_{i}}+\left(\gamma -1\right)\prod_{i=1}^{n}{\left(1-{\left(\mu_{{\overset{˘}{Ҩ}}_{i}}\right)}^{q}\right)}^{{Ⱳ}_{i}}}}\frac{\sqrt[{q}]{\gamma }\prod_{i=1}^{n}{\left({ʋ}_{{\tilde{Ҩ}}_{i}}\right)}^{{Ⱳ}_{i}}}{\sqrt[{q}]{\prod_{i=1}^{n}{\left(1+\left(\gamma -1\right)\left(1-{\left({ʋ}_{{\tilde{Ҩ}}_{i}}\right)}^{q}\right)\right)}^{{Ⱳ}_{i}}+\left(\gamma -1\right)\prod_{i=1}^{n}{\left({ʋ}_{{\tilde{Ҩ}}_{i}}\right)}^{q{Ⱳ}_{i}}}}) \end{array}$$*where *
γ
* > 0, q ≥ 0.*

**Definition** **10.***Darko and Liang presented an operator named the weighted q-rung orthopair fuzzy Hamacher geometric mean (Wq-ROFHGM) [35], as in Equations (26) and (27).**Let*${\overset{˘}{Ҩ}}_{i}$*=*$\left(\mu_{{\tilde{Ҩ}}_{i}},\;{ʋ}_{{\tilde{Ҩ}}_{i}}\right)$* (i = 1, 2, … n) be set of q-ROFNs and**Ⱳ**=*${\left({ⱳ}_{1},{ⱳ}_{2},\;\ldots ,{ⱳ}_{n}\;\right)}^{T}$*be the weight vector of*${\overset{˘}{Ҩ}}_{i}$*with*$\sum_{i=1}^{n}{Ⱳ}_{i}$*= 1 and*${Ⱳ}_{i}$ ∈*[0, 1].*(26)$$\mathrm{Wq}\text{-}\mathrm{ROFHGM}\left({\overset{˘}{Ҩ}}_{1},{\overset{˘}{Ҩ}}_{2},\ldots ,\;{\overset{˘}{Ҩ}}_{n}\right)={ⱳ}_{1}\left({\overset{˘}{Ҩ}}_{1}\right)⊕{ⱳ}_{2}\left({\overset{˘}{Ҩ}}_{2}\right)⊕\ldots \;⊕{ⱳ}_{n}\left({\overset{˘}{Ҩ}}_{n}\right)=\begin{array}{c} n\\ ⊕\\ i=1 \end{array}{ⱳ}_{i}\left({\overset{˘}{Ҩ}}_{i}\right)$$(27)$$\begin{array}{c} \mathrm{Wq}\text{-}\mathrm{ROFHGM}\left({\overset{˘}{Ҩ}}_{1},{\overset{˘}{Ҩ}}_{2},\ldots ,\;{\overset{˘}{Ҩ}}_{n}\right)=\\ \left(\frac{\sqrt[{q}]{\gamma }\prod_{i=1}^{n}{\left(\mu_{{\tilde{Ҩ}}_{i}}\right)}^{{Ⱳ}_{i}}}{\sqrt[{q}]{\prod_{i=1}^{n}{\left(1+\left(\gamma -1\right)\left(1-{\left(\mu_{{\tilde{Ҩ}}_{i}}\right)}^{q}\right)\right)}^{{Ⱳ}_{i}}+\left(\gamma -1\right)\prod_{i=1}^{n}{\left(\mu_{{\tilde{Ҩ}}_{i}}\right)}^{q{Ⱳ}_{i}}}}\sqrt[{q}]{\frac{\prod_{i=1}^{n}{\left(1+\left(\gamma -1\right){\left({ʋ}_{{\overset{˘}{Ҩ}}_{i}}\right)}^{q}\right)}^{{Ⱳ}_{i}}-\prod_{i=1}^{n}{\left(1-{\left({ʋ}_{{\overset{˘}{Ҩ}}_{i}}\right)}^{q}\right)}^{{Ⱳ}_{i}}}{\prod_{i=1}^{n}{\left(1+\left(\gamma -1\right){\left({ʋ}_{{\overset{˘}{Ҩ}}_{i}}\right)}^{q}\right)}^{{Ⱳ}_{i}}+\left(\gamma -1\right)\prod_{i=1}^{n}{\left(1-{\left({ʋ}_{{\overset{˘}{Ҩ}}_{i}}\right)}^{q}\right)}^{{Ⱳ}_{i}}}}\right) \end{array}$$*where *γ* > 0, q ≥ 0.*

### 3.2. Suggested Approach

In this part, a sequential multi-step approach is presented to evaluate several intrusion-detection systems by combining two MCDM methods, namely, Entropy-CoCoSo. The proposed approach was performed under the q-rung orthopair fuzzy environment and by using q-ROFNs. The proposed approach was divided into three main parts. The data aggregation part includes identifying experts, selecting the criteria used, and determining the IDSs alternatives available. Then, the criteria evaluation part assesses the selected criteria using the q-ROF Entropy method. After that, the alternatives evaluation part assesses the available IDSs using the q-ROF CoCoSo method. The steps of the proposed approach are shown in Figure 3. Consequently, the steps of the suggested approach used are presented in detail as follows:

***Step 1*****.** The problem was studied and its main and sub-aspects were identified. The basic criteria for selecting the participating experts also were established. The criteria for selecting experts were determined as follows: the participants should have sufficient experience in the field of cyber security and the field of information security in general; also, their experience in the field of information security should not be less than 10 years. In addition, the participants should have practical experience in the field of information security technology and in the academic field. Next, the number of experts (${\mathrm{EX}}_{s}$) participating in the study was considered. After that, the participating experts were divided into several groups and the appropriate weight for each group was determined according to the measure of experience. Finally, the most appropriate means of communication with the participating experts were determined.

***Step 2***. The main criteria and their sub-aspects used in the study were determined based on an analysis of the relevant literature, as well as insight from the participating experts. $C_{j}$ = $\left\{C_{1},{\;C}_{2},\;\ldots ,{\;C}_{n}\right\}$, with j = 1, 2…n. Let Ⱳ = $\left({ⱳ}_{1},{ⱳ}_{2},\;\ldots ,{ⱳ}_{n}\;\right)$ be the vector set utilized for determining the criteria weights, where ${ⱳ}_{j}$ > 0 and $\sum_{j=1}^{n}{ⱳ}_{j}$ = 1.

***Step 3***. After studying the problem and its details and identifying the most important criteria, the available alternatives were determined to be used in the study. After that, experts’ opinions were taken on the selected alternatives and a final list of alternatives to be used in the evaluation process was prepared. The set $A_{i}$ = $\left\{A_{1},{\;A}_{2},\;\ldots ,{\;A}_{m}\right\}$, having i = 1, 2,..,m alternatives, was evaluated by n decision criteria of set $C_{j}$ = $\left\{C_{1},{\;C}_{2},\;\ldots ,{\;C}_{n}\right\}$, with j = 1, 2, …, n.

***Step 4***. After defining the main criteria and their sub-aspects and adopting a set of final alternatives, all these aspects were organized in the form of a hierarchical structure. This hierarchy shows the main objective of the problem, the criteria used, and the alternatives determined.

***Step 5***.Verbal variants and their equivalent q-ROFNs were identified. These variables were used in the evaluation process to assist the participating experts. These variants were divided into two parts in the same table. The first part refers to the variables that are used in evaluating the main criteria and their sub-aspects. The second part refers to the variables that are used in evaluating the available alternatives, as shown in Table 1.

***Step 6***. Build the q-ROF decision matrices, $G_{\mathrm{EXs}}$, according to experts’ preferences (${\mathrm{EX}}_{s}$) to evaluate the criteria by each expert using verbal variables in Table 1, and then by using q-rung orthopair fuzzy scale q-ROFNs, as shown in Table 2.

***Step 7***. Aggregate the evaluations on criteria weights. The individual expert evaluations are collected by using q−ROFWG given in Equation (23). Here ${\left({ⱳ}_{j}\right)}_{1\times n}$, presents the q-ROF weight of the *j*th criterion.

***Step 8***. The steps of the entropy method based on q-ROFSs are applied to evaluate and weight the main criteria and their sub-aspects [36]. Compute the entropy values of each q-ROFN of the aggregated experts’ evaluations by applying Equations (28) and (29).
(28)$${\mathrm{KE}}_{q,\;\mathrm{ij}}\left(x\right)=\frac{1}{\sqrt{2}}\;\sqrt{{\left({\left(\mu \left(x\right)\right)}^{q}\right)}^{2}+{\left({\left(ʋ\left(x\right)\right)}^{q}\right)}^{2}+{\left({\left(\mu \left(x\right)\right)}^{q}+{\left(ʋ\left(x\right)\right)}^{q}\right)}^{2}}$$
(29)$$\begin{array}{c} {\mathrm{EN}}_{q,\;\mathrm{ij}}\left(x\right)=1-{\mathrm{KE}}_{q,\;\mathrm{ij}}\left(x\right)=1-\\ \frac{1}{\sqrt{2}}\;\sqrt{{\left({\left(\mu \left(x\right)\right)}^{q}\right)}^{2}+{\left({\left(ʋ\left(x\right)\right)}^{q}\right)}^{2}+{\left({\left(\mu \left(x\right)\right)}^{q}+{\left(ʋ\left(x\right)\right)}^{q}\right)}^{2}} \end{array}$$

***Step 9***. The main criteria weights are calculated based on the entropy values using Equation (30).
(30)$${ⱳ}_{j}=\frac{1-{ℇ}_{j}}{\sum_{j=1}^{n}\left(1-{ℇ}_{j}\right)};\;j=1,\;2,\ldots \;n\;$$
where ${ℇ}_{j}$ = $\frac{\sum_{i=1}^{m}{\mathrm{EN}}_{q,\;\mathrm{ij}}}{\sum_{i=1}^{m}\sum_{j=1}^{n}.{\mathrm{EN}}_{q,\;\mathrm{ij}}}$ refers to the q-ROF entropy value.

***Step 10***. In the same way, the weights of the sub-aspects of the main criteria are calculated as in Steps 6–9.

***Step 11***. A q-ROF evaluation decision matrix ($T_{EX}$) is generated by each expert (*EX*) individually among the selected sub-aspects and alternatives to determine the best intrusion-detection system through the use of verbal variables, as shown in Table 1, and then by using the q-ROFNs in Table 1, as shown in Table 3. Here, ${\overset{˘}{T}}_{EX}$ = ${\left({\overset{˘}{t}}_{ijEX}\right)}_{n\times m}$, in which ${\overset{˘}{t}}_{ijEX}$ = $\left[\mu_{ijEX}\;,{ʋ}_{ijEX}\right]$ is created by applying the verbal variables in Table 1. Consequently, ${\overset{˘}{t}}_{ijEX}$ refers the performance of intrusion-detection systems (alternatives) $A_{i}$ according to criteria $C_{j}$ of the ${\mathrm{EX}}^{\mathrm{th}}$ expert.

***Step 12***. After the q-ROF evaluation decision matrix ($T_{EX}$) is generated by each expert (*EX*) between the sub-aspects and the available alternatives by all experts, the q-ROF evaluation decision matrices ($T_{EXs}$) were aggregated into one matrix by utilizing q−ROFWG, as presented in Equation (23). A combined q-ROF evaluation decision matrix ($\overset{˘}{T}$) was created as in Table 4. Accordingly, $\overset{˘}{T}$ = ${\left({\overset{˘}{t}}_{ij}\right)}_{n\times m}$ in which ${\overset{˘}{t}}_{ij}$ = $\left[\mu_{ij}\;,{ʋ}_{ij}\right]$ is utilized to refer to the combined q-ROFN of the *i*th substitute with regard to the *j*th criteria.

***Step 13***. Compute the normalized aggregated q-ROF evaluation decision matrix ($\overset{˘}{H}$) by applying Equation (31).
(31)$$\overset{˘}{H}={\left({\overset{˘}{h}}_{ij}\right)}_{m\times n}=\left[{\overset{˘}{\mu }}_{ij}\;,{\overset{˘}{ʋ}}_{ij}\right]=\{\begin{array}{c} \left({\overset{˘}{\mu }}_{ij}\;,{\overset{˘}{ʋ}}_{ij}\right)\\ \left({\overset{˘}{ʋ}}_{ij}\;,{\overset{˘}{\mu }}_{ij}\right) \end{array}\begin{array}{c} if\;i\;\in Ƀ\\ if\;i\;\in Ȼ \end{array}$$
where Ƀ refers to the set of benefit criteria and Ȼ refers to the set of cost criteria.

***Step 14***. Calculate the total of the weighted comparability arrangement (ἁ) for all substitutes by applying the Wq-ROFHA operator as presented in Equations (24) and (25).

***Step 15***. Calculate the full of the power weight (ẞ) of comparability arrangements for all substitutes by applying the Wq-ROFHGM operator as exhibited in Equations (26) and (27).

***Step 16***. Determine the score values of the substitutes by applying the values of values of the Wq-ROFHA and Wq-ROFHGM operators for each substitute by applying Equation (20).

***Step 17***. Calculate the proportional weight of the substitutes with the assistance of Equations (32)–(34).
(32)$${Ɣ}_{ja}\;=\frac{\;{ἁ}_{j}+{ẞ}_{j}}{\sum_{j=1}^{n}\left({ἁ}_{j}+{ẞ}_{j}\right)}$$(33)$${Ɣ}_{jb}\;=\frac{\;{ἁ}_{j}}{\min\left({ἁ}_{j}\right)}+\frac{{ẞ}_{j}}{\min\left({ẞ}_{j}\right)}$$(34)$${Ɣ}_{jc}=\frac{Ѱ{ἁ}_{j}+\left(1-Ѱ\right){ẞ}_{j}}{Ѱ\max\left({ἁ}_{j}\right)+\left(1-Ѱ\right)\max\left({ẞ}_{j}\right)\;};0\leq Ѱ\leq 1$$
where $\;{Ɣ}_{ja}$ refers to the arithmetic mean of sums of the weighted sum method (WSM) and weighted product model (WPM) scores. Then, ${Ɣ}_{jb}$ indicates the sum of proportional scores of WSM and WPM. ${Ɣ}_{jc}$ also refers to the stable adjustment of the WSM and WPM models scores.

***Step 18***. Determine the evaluation values (${Ɣ}_{j}$) of the substitutes by applying Equation (35). Then, rank the available intrusion-detection systems according to the most possible value of the evaluation values (${Ɣ}_{j}$).
(35)$${Ɣ}_{j}=\sqrt[{3}]{{Ɣ}_{ja}{Ɣ}_{jb}{Ɣ}_{jc}}+\frac{{Ɣ}_{ja}+{Ɣ}_{jb}+{Ɣ}_{jc}}{3}$$

## 4. Empirical Results and Interpretation

In this section, the steps of the proposed multistep hybrid MCDM approach consisting of Entropy-CoCoSo methods are applied to evaluate the efficiency and reliability of some IDSs. The proposed approach was applied under the q-rung orthopair fuzzy environment and by using q-ROFNs. In this regard, the process of evaluating intrusion-detection systems and selecting the most effective and reliable one is necessary and inevitable in light of the recent cyber-attacks and intrusion methods. In this regard, the intrusion-detection systems are revealed in the next sub-section.

### 4.1. Description of Intrusion-Detection Systems

In this subsection, a brief description of the intrusion-detection systems are considered; also, Figure 4 demonstrates the general structure of the network and IDS.

**Suricata (**$IDS_{1}$**):** Suricata was developed by the Open Information Security Foundation in 2010. Suricata is the main alternative to Snort because the design of Suricata is very close to that of Snort [37]. Suricata has an advantage over Snort, which is that it collects data at the application layer. Suricata consists of so-called threads, thread units, and queues. Suricata is a multi-threaded program, so there will be multiple threads running at the same time [37]. Thread units are divided according to functions; for example, one unit is used to analyze data packets, and the other unit is used to discover data packets. Each data packet can be processed by several different threads, and the queue is used to transfer the data packet from one thread to another. At the same time, a thread can contain several thread units, but only one unit runs at a given time.**Zeek (**$IDS_{2}$**):** Zeek (previously known as Bro until 2019) is a network intrusion-detection system that is compatible with Linux, Unix, and Mac OS [38]. Zeek uses network-based intrusion-detection methods by tracking the network and searching for malicious activities. The Zeek intrusion detection function is realized in two stages: traffic logging, and analysis. As with Suricata, Zeek has a significant advantage over Snort in that its analysis runs at the application layer, resulting in a broader analysis of network protocol activity.**Security onion (**$IDS_{3}$**):** Security Onion is a Linux-based IDS that is a mixture of several IDS that are both HIDS and NIDS solutions [16]. Although Security Onion is classified as NIDS, it includes HIDS functionality as well. It monitors log and configuration files for suspicious activity and checks those files for any unexpected changes. One of the downsides to Security Onion’s comprehensive network monitoring system is its complexity. Thus, the Security Onion analysis engine is where things get complicated because there are so many different tools with different operating procedures that most of them may end up being overlooked.**Snort (**$IDS_{4}$**):** Snort is a Linux-based lightweight cross-platform network intrusion-detection system that can be used to monitor TCP/IP networks [39]. Snort is easy to deploy and can be configured to monitor network traffic for intrusion attempts, log intrusion behavior, and perform specific actions when intrusion attempts are detected. It is one of the most widely deployed IDS tools and can also be used as an intrusion-prevention system. Snort can be traced back to 1998, and there are still no signs of disappearing. There are some active communities that offer good help and support. The high level of personalization that Snort provides makes it a good choice for many different organizations.**Wazuh (**$IDS_{5}$**):** Wazuh is an IDS used to detect security and monitor compliance with security rules. Wazuh is an open-source intrusion-detection system project. It was developed as a fork part of OSSEC HIDS and was later integrated with Elastic Stack and Opens CAP. It relies on a cross-platform approach that redirects system data such as log messages, file tables, and detected anomalies to a central manager, where it is further analyzed and processed, resulting in security alerts. It monitors the file system and identifies changes in content, permissions, ownership, and file properties that need to be monitored. It monitors configuration files to ensure that they comply with security policies, standards, or hardening guides.**OSSEC (**$IDS_{6}$**):** OSSEC is an open source IDS developed by Daniel B. Cid, who had sold the system to Trend Micro in 2008 [39]. Its detection methods are based on checking log files, making it a host-based IDS. OSSEC works on Unix, Linux, Mac OS, and Windows. There is no front end for this tool, but you can connect with Kibana or Graylog. OSSEC disclosure rules are called ‘Policies’. You can write your own policies or get packages of them for free from the user community. It is also possible to define actions that should be performed automatically when specific warnings appear.

### 4.2. Application of the Proposed Approach

In this sub-section, the steps for evaluating the selected intrusion-detection systems through the Entropy-CoCoSo approach are presented as follows: 

***Step 1***. Initially, a set of standards was identified to select the experts involved with the researchers in the study to evaluate the IDSs. The standards were as follows: the number of years of experience should not be less than 10 years in the field of cyber security and the field of information security in general; also, the scientific degree of the participating experts must not be lower than M.Sc. Accordingly, 60 experts were selected to participate in the IDSs evaluation process. After that, the participating experts were divided into four groups. Each group included a certain number of experts. The first and fourth groups included 12 experts. Whereas, the second and third groups included 18 experts. Accordingly, the appropriate weight was assigned to each group according to the years of experience and the number of experts. So, the first, second, third, and fourth groups had weights of 0.20, 0.30, 0.30, and 0.20, respectively. In addition, a leader was assigned to each group to express the final opinion in the evaluation process. Finally, the experts were contacted online.

***Step 2***. Based on the literature analysis and expert review, a set of main and sub-criteria were defined to evaluate the effectiveness and reliability of the IDSs. Initially, four main criteria were defined, which are as follows: protected system, ${\mathrm{PSC}}_{1}$; audit source location, ${\mathrm{ASC}}_{2}$; targets, ${\mathrm{TGC}}_{3}$; and types, ${\mathrm{TPC}}_{4}$. The main criteria also contained several sub-criteria, as follows: HIDS (${\mathrm{HIC}}_{1\_1}$), NIDS (${\mathrm{NIC}}_{1\_2}$), hybrids (${\mathrm{HYC}}_{1\_3}$), host log files (${\mathrm{HLC}}_{2\_1}$), network packets (${\mathrm{NPC}}_{2\_2}$), application log files (${\mathrm{ALC}}_{2\_3}$), IDS sensors alerts (${\mathrm{ISC}}_{2\_4}$), applications (${\mathrm{APC}}_{3\_1}$), network (${\mathrm{NEC}}_{3\_2}$), host (${\mathrm{HOC}}_{3\_3}$), open source (${\mathrm{OSC}}_{4\_1}$), closed source (${\mathrm{CSC}}_{4\_2}$), and freeware (${\mathrm{CSC}}_{4\_2}$).

***Step 3***. A definitive list of available IDSs for use was prepared, as follows: Suricata (${\mathrm{IDS}}_{1}$), Zeek (${\mathrm{IDS}}_{2}$), Security onion (${\mathrm{IDS}}_{3}$), Snort (${\mathrm{IDS}}_{4}$), Wazuh (${\mathrm{IDS}}_{5}$), and OSSEC (${\mathrm{IDS}}_{6}$).

***Step 4***. A final hierarchical form of the problem was prepared, defining the main objective of the study, which was to evaluate the effectiveness of several IDSs, as shown in Figure 5; this in addition to regulating the relationship between the basic criteria and their sub-aspects, with the IDSs used as alternatives.

***Step 5***. A set of verbal variants and their equivalent q-ROFNs were prepared by reviewing the previous literature and expert opinions. Verbal variants were divided into two parts. The first part of the verbal variants is presented in Table 1, to assess the main criteria and their sub-aspects. The second part of the verbal variants is presented in Table 1, to evaluate the alternatives used.

***Step 6***. The decision matrix was built with the help of Table 2 by the four experts to assess the main criteria using the verbal variables as shown in Table 5.

***Step 7***. Individual expert evaluations of the main criteria were compiled using a q-ROFWG operator in Equation (23), and using the weights assigned to the four experts, namely, 0.2, 0.3, 0.3, and 0.2, respectively, as exhibited in Table 5. The parameter was determined in a discretionary manner to reflect the position of the experts in terms of optimism and pessimism. In this case, q = 5 was introduced for a stronger illustration of the uncertainty.

***Step 8.*** The entropy method was applied to calculate the entropy values for each q-ROFN from the aggregated experts’ evaluations by applying Equations (28) and (29), as shown in Table 5.

***Step 9.*** The weights of the main criteria were calculated based on the entropy values using Equation (30), as presented in Table 5.

***Step 10.*** Likewise, the weights of the sub-aspects of the main criteria were calculated, as presented in Table 6, Table 7, Table 8 and Table 9. Accordingly, the global weights of the sub-aspects were calculated, as in Table 10.

***Step 11.*** An evaluation decision matrix was established to evaluate the IDSs according to the sub-aspects by the four experts and with the assistance of Table 3, as presented in Table 11.

***Step 12***. Individual expert evaluations of the alternatives between the sub-aspects and the available alternatives were compiled using the q-ROFWG operator in Equation (23), and using the weights assigned to the four experts, namely, 0.2, 0.3, 0.3, and 0.2, respectively, as exhibited in Table 12. The parameter also was determined in a discretionary manner to reflect the position of the experts in terms of optimism and pessimism. In this case, q = 5 and γ =1 were introduced for a stronger illustration of the uncertainty.

***Step 13***. The normalized aggregated q-ROF evaluation decision matrix was calculated by applying Equation (31), as presented in Table 13.

***Step 14***. Calculate the total of the weighted comparability arrangement for all alternatives by applying the Wq-ROFHA operator, as presented in Equations (24) and (25), and as exhibited in Table 14.

***Step 15***. Calculate the full power weight of the comparability arrangements for all alternatives by applying the Wq-ROFHGM operator, as exhibited in Equations (26) and (27), and as exhibited in Table 14.

***Step 16***. The score values of the all alternatives are computed by applying the values of the Wq-ROFHA and Wq-ROFHGM operators for each alternative by applying Equation (20), as presented in Table 15.

***Step 17***. The proportional weight of the alternatives is calculated with the assistance of Equations (32)–(34), as shown in Table 16.

***Step 18***. The evaluation values (${Ɣ}_{j}$) of the alternatives are identified by applying Equation (35). Then, the six intrusion-detection systems are rated according to the most possible value of the evaluation values (${Ɣ}_{j}$), as presented in Table 16 and in Figure 6.

### 4.3. Results Interpretation

In this subsection, some interpretations are introduced of the results obtained from applying the proposed approach, Entropy-CoCoSo, under a q-rung orthopair fuzzy environment. The results obtained are divided into two parts. The first part relates to the results of the main criteria weights and their sub-aspects. The second part relates to the results of the intrusion-detection systems evaluation used in the study. Initially, the four main criteria were evaluated by the participating experts. The results obtained indicate that the ${\mathrm{PSC}}_{1}$ criterion has the highest weight, with a weight of 0.278, followed by the ${\mathrm{TGC}}_{3}$ criterion with a weight of 0.266. Whereas, the ${\mathrm{ASC}}_{2}$ criterion has the lowest weight, 0.226, and occupies the last rank in the ranking of the main criteria. Accordingly, the sub-criteria related to each main criterion were evaluated. Thus, the sub-criteria related to the ${\mathrm{PSC}}_{1}$ criterion were evaluated as follows: the ${\mathrm{NIC}}_{1\_2}$ criterion has the top weight with a weight of 0.377, followed by the ${\mathrm{HIC}}_{1\_1}$ criterion with a weight of 0.319; the ${\mathrm{HYC}}_{1\_3}$ criterion has the lowest weight, 0.304. The sub-criteria related to the ${\mathrm{ASC}}_{2}$ criterion were calculated as follows: the ${\mathrm{HLC}}_{2\_1}$ criterion has the maximum weight, with a weight of 0.282, followed by the ${\mathrm{ISC}}_{2\_4}$ criterion, with a weight of 0.271; the ${\mathrm{ALC}}_{2\_3}$ criterion has the minimum weight, 0.196. In addition, the sub-criteria related to the ${\mathrm{TGC}}_{3}$ criterion were estimated as follows: the ${\mathrm{NEC}}_{3\_2}$ criterion has the highest weight, with a weight of 0.404, followed by the ${\mathrm{HOC}}_{3\_3}$ criterion with a weight of 0.312; the ${\mathrm{APC}}_{3\_1}$ criterion has the lowest weight, 0.284. Afterward, the sub-criteria related to the ${\mathrm{TPC}}_{4}$ criterion were assessed as follows: the ${\mathrm{FRC}}_{4\_3}$ criterion has the largest weight, with a weight of 0.379, followed by the ${\mathrm{OSC}}_{4\_1}$ criterion with a weight of 0.322; the ${\mathrm{CSC}}_{4\_2}$ criterion has the smallest weight, 0.299.

In the end, the results of the intrusion-detection systems used in the evaluation process were revealed as follows: Suricata (${\mathrm{IDS}}_{1}$) has the top rank with a weight of 2.398 followed by Snort (${\mathrm{IDS}}_{4}$) with a weight of 2.089. In turn, Zeek (${\mathrm{IDS}}_{2}$) has the lowest rank with a weight of 1.595.

### 4.4. Comparative Analysis

In this sub-section, a comparative analysis is demonstrated to test and verify the effectiveness of the developed approach, q-ROF Entropy-CoCoSo. Consequently, the assessment results were compared with Alyami et al.’s [16] fuzzy AHP–TOPSIS approach. In this regard, the same weights of the main criteria and sub-aspects obtained by applying the proposed approach were used, as shown in Table 10. Accordingly, the results of ranking the alternatives used in the study using the two approaches are presented in Table 17 and in Figure 7. The results of the comparison show that ${\mathrm{IDS}}_{1}$ is the best alternative according to the results of the two approaches. ${\mathrm{IDS}}_{2}$ is the least alternative in the order. According to the results, it can be seen that there are some changes in the order of some alternatives, such as ${\mathrm{IDS}}_{3}$, ${\mathrm{IDS}}_{4}$, ${\mathrm{IDS}}_{5}$, and ${\mathrm{IDS}}_{6}$. The presence of some differences in the order of the alternatives can be explained by the difference in the mathematical basis for each approach. Finally, the results of the comparative analysis and the reliability of the proposed approach can be verified by the experts.

### 4.5. Sensitivity Analysis

We have conducted a sensitivity analysis from the three perspectives of changes in parameter q, parameter γ, and parameter Ѱ. Sensitivity analysis was conducted on the results obtained to confirm their reliability and stability and to examine the change that occurred to them as a result of the change in some inputs and parameters. In decision-making approaches, some parameters are defined subjectively based on the perception of the problem by decision-makers and the extent of the risks in the environment. Consequently, these parameters change according to the circumstances in which the decision-making system is being modeled. In our proposed Entropy-CoCoSo q-ROF approach, three parameters—q, γ, and Ѱ—are defined, which are determined based on the personal preferences of the experts. Accordingly, several changes were made to these parameters to show their decisive influence on the final IDS’s ranking results. These changes were divided into four scenarios. The first scenario refers to the change in the values of parameter q. The second scenario indicates the change in the values of parameter γ. The third scenario refers to the change in the values of parameters q and γ. Lastly, the fourth scenario refers to the change in the values of parameter Ѱ.

The first scenario is the effect of a change in parameter q on the evaluation of IDSs. Accordingly, the value of the parameter q was changed several times, from q = 2 to q = 20, to show its impact on the evaluation of IDSs, as presented in Figure 8. Although the value of the q parameter has been changed several times, the order of the IDSs has not changed at all. ${\mathrm{IDS}}_{1}$ remains the best alternative throughout the sensitivity analysis and parameter value change q, followed by ${\mathrm{IDS}}_{4}$. By contrast, ${\mathrm{IDS}}_{2}$ remains the lowest in order despite the change in the value of the parameter q. The changes that can be observed based on the change in the value of the parameter q in the order of the IDSs, show there is a large convergence between the values of the assessment of ${\mathrm{IDS}}_{4}$ and ${\mathrm{IDS}}_{6}$ at the value of q = 2. Significant convergence occurs between the ${\mathrm{IDS}}_{4}$ and ${\mathrm{IDS}}_{1}$ assessment values at q = 8; otherwise, the order of the IDSs remains the same despite the presence of some increases in the weights of the IDSs.

The second scenario is the effect of a change in parameter γ on the evaluation of IDSs. Accordingly, the value of the parameter γ was changed several times, from γ = 0.1 to γ = 1.0, to show its effect on the evaluation of IDSs, as shown in Figure 9. Although the value of the parameter γ was changed several times, the order of the IDSs changed only when the value of parameter γ = 1.0. ${\mathrm{IDS}}_{1}$ remains the best alternative throughout the sensitivity analysis and parameter value change γ = 0.1 to γ = 0.9 followed by ${\mathrm{IDS}}_{4}$, except when parameter value γ = 1.0, then ${\mathrm{IDS}}_{6}$ becomes the second rank in the analysis process. In contrast,${\;\mathrm{IDS}}_{2}$ remains lowest in order throughout the change of the value of the parameter γ = 0.0 to γ = 0.9, except when the value of γ = 1.0 is changed, then ${\mathrm{IDS}}_{2}$ becomes the fifth rank, penultimate. The changes that can be observed based on the change in the value of the parameter q in the order of IDSs, is that when the value of the parameter γ = 1.0, the order of the IDSs changes so that ${\mathrm{IDS}}_{1}$ is in the first order, while ${\mathrm{IDS}}_{6}$ is in the second order, and ${\mathrm{IDS}}_{4}$ is in the third order. On the contrary, the rank of some IDSs was changed, such as the rank of ${\mathrm{IDS}}_{2}$ and the ${\mathrm{IDS}}_{5}$, which became the fifth and sixth, respectively.

The third scenario is the effect of a change in parameter q and γ on the evaluation of IDSs. Accordingly, the values of q and γ were changed several times, from q = 2 to q = 15, and γ = 0.1 to γ = 1 to show their combined effect on the assessment of the IDSs, as shown in Figure 10. Although the values of the parameters q and γ changed many times, the order of the IDSs has not changed at all. ${\mathrm{IDS}}_{1}$ remains the best alternative during sensitivity analysis and changing the values of q and γ parameters, followed by ${\mathrm{IDS}}_{4}$. In contrast, ${\mathrm{IDS}}_{2}$ remains the lowest in terms of rank despite the change in the values of the two parameters q and γ. Changes that can be observed based on the change in the value of parameters q and γ for the order of IDSs, is that there is a great convergence between the values of the evaluation process weights for the IDSs used in the study. The convergence between the weights of the IDSs is difficult to see in Figure 10, and this is one of the shortcomings of Figure 10.

The fourth scenario is the effect of a change in parameter Ѱ on the evaluation of IDSs. Accordingly, the value of parameter Ѱ was changed several times from Ѱ = 0.1 to Ѱ = 1.0, to show its effect on the evaluation of the IDSs, as shown in Figure 11. Although the value of parameter Ѱ was changed several times, the order of the IDSs did not change at all. ${\mathrm{IDS}}_{1}$ remains the best alternative throughout the sensitivity analysis and parameter value change Ѱ = 0.1 to Ѱ = 0.9, followed by ${\mathrm{IDS}}_{4}$. On the contrary, ${\mathrm{IDS}}_{2}$ remains in the lowest order by changing the parameter value Ѱ = 0.1 to Ѱ = 1.0.

## 5. Conclusions

Given the spread of computer networks and the dependence of public and private institutions on their efficiency and quality of work, any disruption or sabotage of them may lead to great losses. Information systems and networks are constantly subject to cyber-attacks. Thus, firewalls and antivirus are not enough to fend off all these attacks, as they are only able to protect the “front entrance” of computer systems and networks. IDSs can help protect your corporation from malicious activities. There are different types of IDSs to protect networks, as intrusion attacks are becoming more and more common on a global scale. In addition, hackers using new technologies are trying to penetrate systems. An IDS is a tool that identifies these attacks and will take an immediate step to get the system back to normal, as the IDS can also detect network traffic and send an alarm if a breach is found.

In this regard, this study discusses the most effective and used IDSs. This study was conducted with the participation of many experts under the q-ROF environment to deal with the uncertainty that may occur as a result of different circumstances and differences in evaluation frameworks. Six intrusion-detection systems, namely, Suricata (${\mathrm{IDS}}_{1}$), Zeek (${\mathrm{IDS}}_{2}$), Security onion (${\mathrm{IDS}}_{3}$), Snort (${\mathrm{IDS}}_{4}$), Wazuh (${\mathrm{IDS}}_{5}$), and OSSEC (${\mathrm{IDS}}_{6}$), were evaluated according to four key criteria and thirteen sub-aspects. The main criteria were protected system, audit source location, targets, and types. The sub-aspects, on the basis of which the effectiveness of the intrusion-detection systems was evaluated, were HIDS, NIDS, hybrids, host log files, network packets, application log files, IDS sensors alerts, applications, network, host, open-source, closed source, and freeware. A hybrid MCDM approach, including q-ROF entropy-CoCoSo techniques, was proposed, where entropy was applied to evaluate the main criteria and their sub-aspects. The CoCoSo method is applied to rate six IDSs according to their effectiveness. Afterward, comparative and sensitivity analyses were performed to confirm the stability, reliability, and performance of the proposed approach. The findings indicate that most of the IDSs appear to be systems with high potential. According to the results, Suricata is the best IDS that relies on multi-threading performance. Although the results here confirm that the proposed method is applicable and effective, it has some limitations. The key limitation of the approach is the difficult mathematical algorithm for the computation of Hamacher functions.

## Figures and Tables

**Figure 2 sensors-22-04123-f002:**
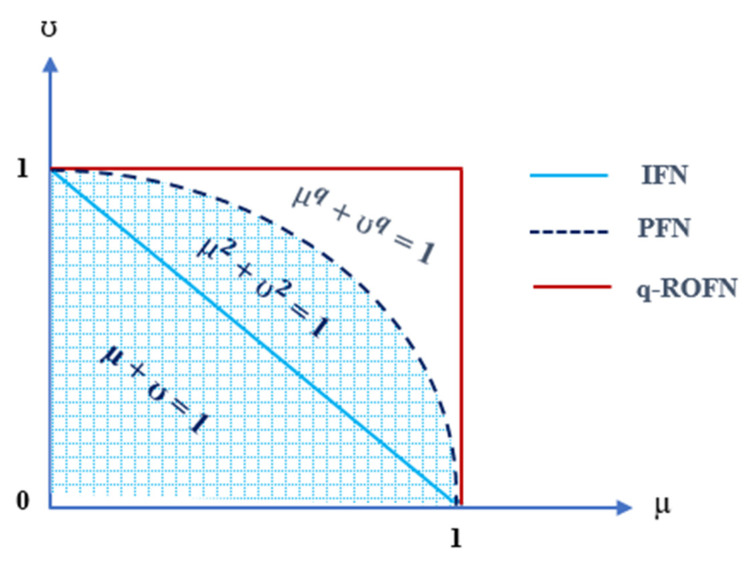
Comparison of the geometric area of various fuzzy membership degrees: IFNs, PFNs, and q-ROFNs.

**Figure 3 sensors-22-04123-f003:**
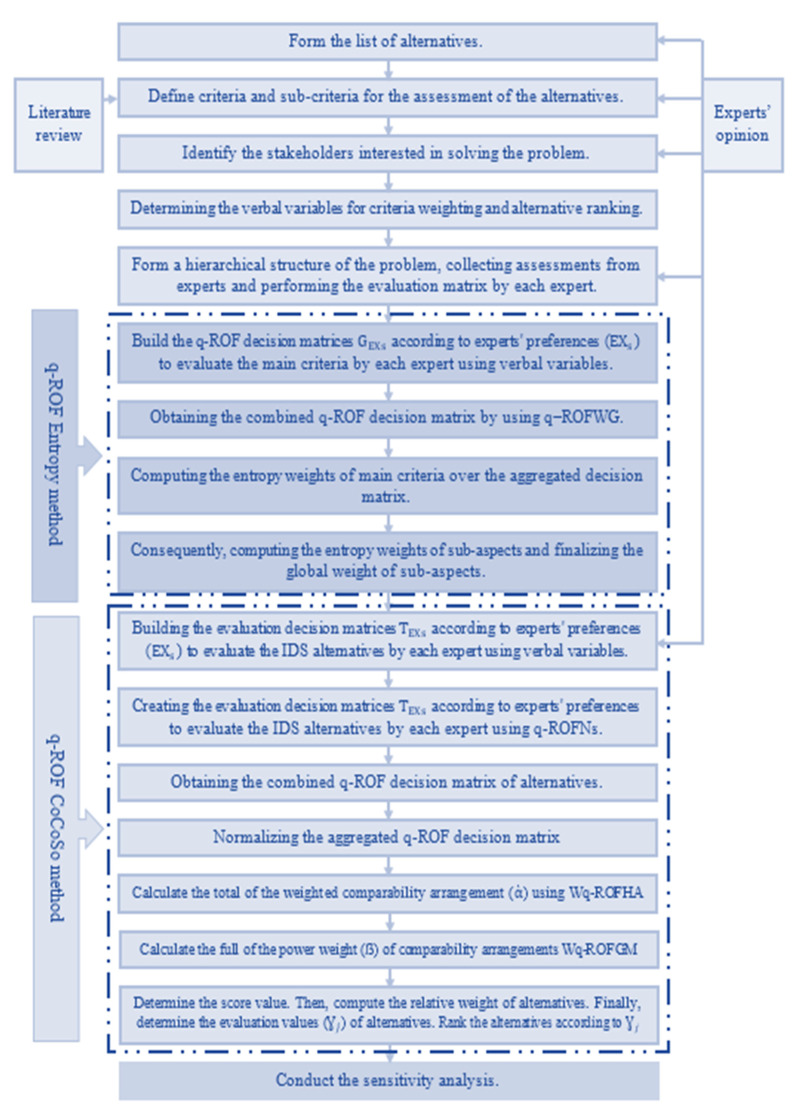
Decision framework for IDS evaluation.

**Figure 4 sensors-22-04123-f004:**
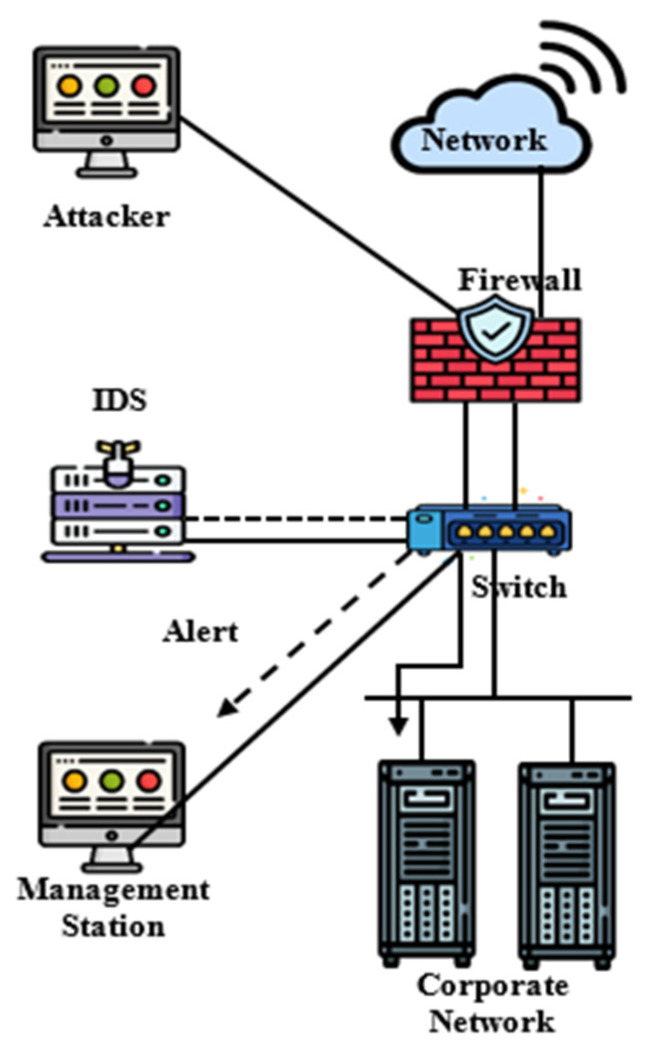
The general structure of the network and IDS.

**Figure 5 sensors-22-04123-f005:**
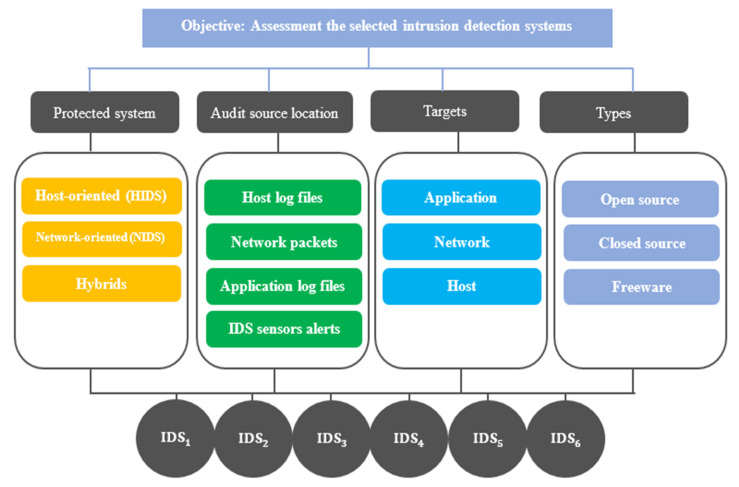
The hierarchy structure of the problem.

**Figure 6 sensors-22-04123-f006:**
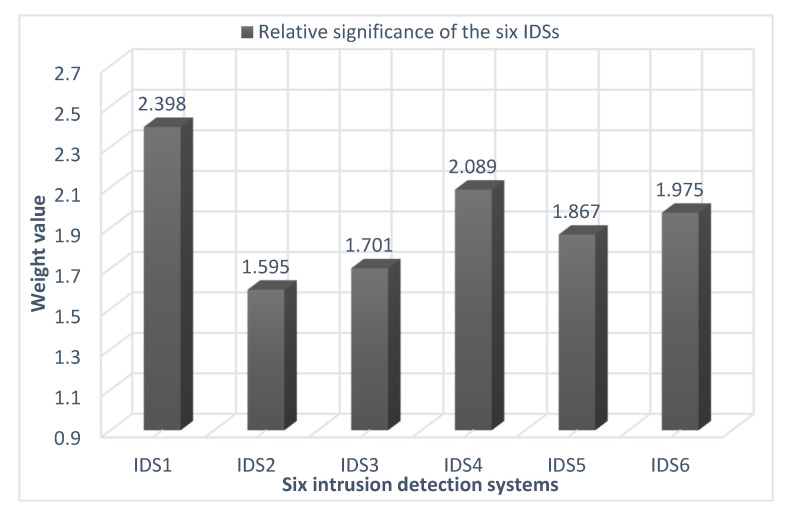
Ranking of the IDSs using the CoCoSo method.

**Figure 7 sensors-22-04123-f007:**
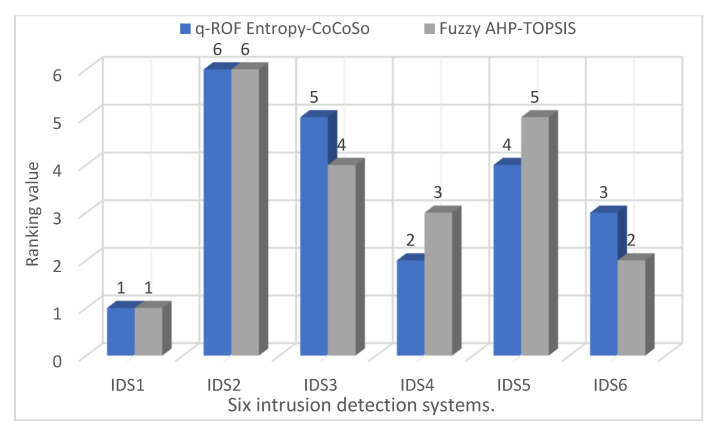
Final ranking of the six IDSs using various approaches.

**Figure 8 sensors-22-04123-f008:**
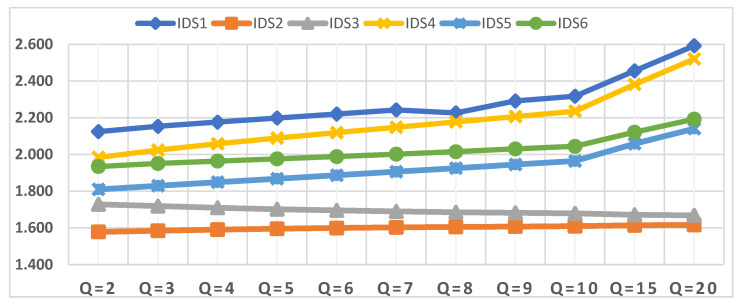
Closeness coefficient values of IDSs in terms of different values of q.

**Figure 9 sensors-22-04123-f009:**
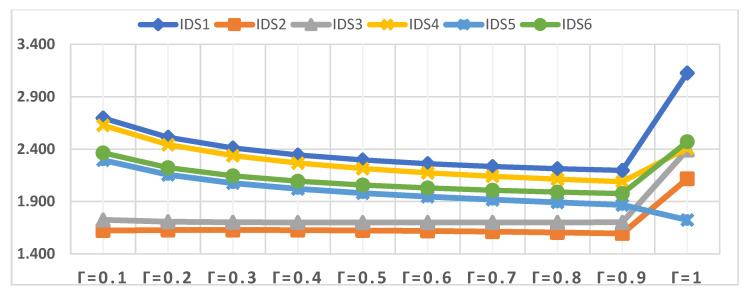
Closeness coefficient values of IDSs in terms of different values of γ.

**Figure 10 sensors-22-04123-f010:**
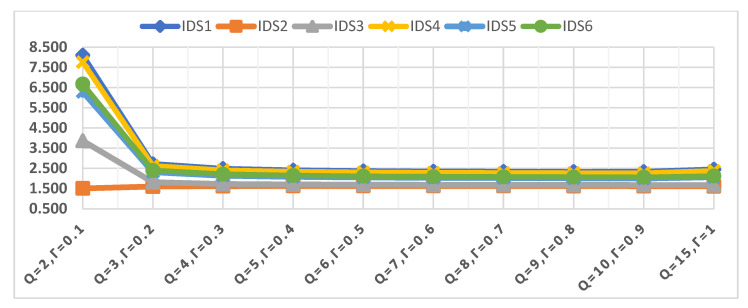
Closeness coefficient values of IDSs in terms of the different values of q and γ.

**Figure 11 sensors-22-04123-f011:**
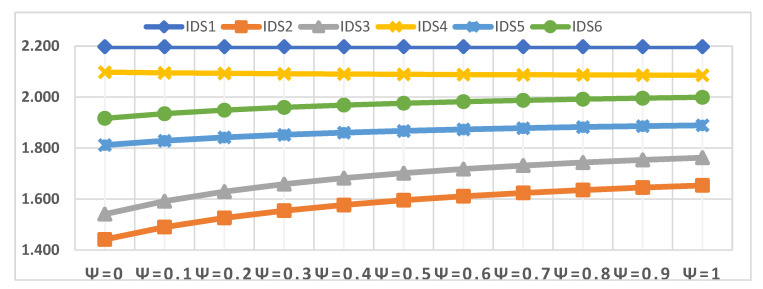
Closeness coefficient values of the IDSs in terms of the different values of Ѱ.

**Table 1 sensors-22-04123-t001:** Verbal variables and their corresponding q-ROFNs for the weighting criteria and ranking alternatives.

Verbal Variables for Criteria	Abbreviations for Criteria	Verbal Variables for Alternatives	Abbreviations for Alternatives	q-ROFNs	
				μ	ʋ
Extremely poor	ELP	Extremely low	EXO	0.11	0.99
Very poor	VPO	Very low	VLO	0.22	0.88
Poor	POO	Low	LLO	0.33	0.77
Medium poor	MDP	Medium low	MEL	0.44	0.66
Fair	FAR	Medium	MED	0.55	0.55
Medium good	MDG	Medium high	MEH	0.66	0.44
Good	GOO	High	HGH	0.77	0.33
Very good	VGO	Very high	VEH	0.88	0.22
Extremely good	EXG	Extremely high	EXH	0.99	0.11

**Table 2 sensors-22-04123-t002:** The evaluation matrix for criteria based on q-ROFN with respect to experts.

Criteria	Experts			
	$Ex_{1}$	$Ex_{2}$	$Ex_{3}$	$Ex_{4}$
$C_{1}$	$\left[\mu_{1{\mathrm{Ex}}_{1}}\;,{ʋ}_{1{\mathrm{Ex}}_{1}}\right]$	$\left[\mu_{1{\mathrm{Ex}}_{2}}\;,{ʋ}_{1{\mathrm{Ex}}_{2}}\right]$	$\left[\mu_{1{\mathrm{Ex}}_{3}}\;,{ʋ}_{1{\mathrm{Ex}}_{3}}\right]$	$\left[\mu_{1{\mathrm{Ex}}_{4}}\;,{ʋ}_{1{\mathrm{Ex}}_{4}}\right]$
$C_{2}$	$\left[\mu_{2{\mathrm{Ex}}_{1}}\;,{ʋ}_{2{\mathrm{Ex}}_{1}}\right]$	$\left[\mu_{2{\mathrm{Ex}}_{2}}\;,{ʋ}_{2{\mathrm{Ex}}_{2}}\right]$	$\left[\mu_{2{\mathrm{Ex}}_{3}}\;,{ʋ}_{2{\mathrm{Ex}}_{3}}\right]$	$\left[\mu_{2{\mathrm{Ex}}_{4}}\;,{ʋ}_{2{\mathrm{Ex}}_{4}}\right]$
⋮	⋮	⋮	⋮	⋮
$C_{n}$	$\left[\mu_{n{\mathrm{Ex}}_{1}}\;,{ʋ}_{{\mathrm{nEx}}_{1}}\right]$	$\left[\mu_{{\mathrm{nEx}}_{2}}\;,{ʋ}_{{\mathrm{nEx}}_{2}}\right]$	$\left[\mu_{{\mathrm{nEx}}_{3}}\;,{ʋ}_{n{\mathrm{Ex}}_{3}}\right]$	$\left[\mu_{{\mathrm{nEx}}_{4}}\;,{ʋ}_{{\mathrm{nEx}}_{4}}\right]$

**Table 3 sensors-22-04123-t003:** Decision evaluation matrix for alternatives in terms of criteria based on q-ROFN.

Criteria	Alternatives (Intrusion-Detection Systems)			
	$A_{1}$	$A_{2}$	…	$A_{m}$
$C_{1}$	$\left[\mu_{11\mathrm{Ex}}\;,{ʋ}_{11\mathrm{Ex}}\right]$	$\left[\mu_{12\mathrm{Ex}}\;,{ʋ}_{12\mathrm{Ex}}\right]$	…	$\left[\mu_{1\mathrm{mEx}}\;,{ʋ}_{1\mathrm{mEx}}\right]$
$C_{2}$	$\left[\mu_{21\mathrm{Ex}}\;,{ʋ}_{21\mathrm{Ex}}\right]$	$\left[\mu_{22\mathrm{Ex}}\;,{ʋ}_{22\mathrm{Ex}}\right]$	…	$\left[\mu_{2\mathrm{mEx}}\;,{ʋ}_{2\mathrm{mEx}}\right]$
⋮	⋮	⋮	⋱	⋮
$C_{n}$	$\left[\mu_{n1\mathrm{Ex}}\;,{ʋ}_{n1\mathrm{Ex}}\right]$	$\left[\mu_{n2\mathrm{Ex}}\;,{ʋ}_{n2\mathrm{Ex}}\right]$	…	$\left[\mu_{\mathrm{nmEx}}\;,{ʋ}_{\mathrm{nmEx}}\right]$

**Table 4 sensors-22-04123-t004:** Combined evaluation matrix for alternatives in terms of the criteria based on q-ROFN.

Criteria	Alternatives (Intrusion-Detection Systems)			
	$A_{1}$	$A_{2}$	…	$A_{m}$
$C_{1}$	$\left[\mu_{11}\;,{ʋ}_{11}\right]$	$\left[\mu_{12}\;,{ʋ}_{12}\right]$	…	$\left[\mu_{1m}\;,{ʋ}_{1m}\right]$
$C_{2}$	$\left[\mu_{21}\;,{ʋ}_{21}\right]$	$\left[\mu_{22}\;,{ʋ}_{22}\right]$	…	$\left[\mu_{2m}\;,{ʋ}_{2m}\right]$
⋮	⋮	⋮	⋱	⋮
$C_{n}$	$\left[\mu_{n1}\;,{ʋ}_{n1}\right]$	$\left[\mu_{n2}\;,{ʋ}_{n2}\right]$	…	$\left[\mu_{\mathrm{nm}}\;,{ʋ}_{\mathrm{nm}}\right]$

**Table 5 sensors-22-04123-t005:** Verbal evaluations of the main criteria by each expert and the aggregated main criteria weights.

Main Criteria	$Ex_{1}$	$Ex_{2}$	$Ex_{3}$	$Ex_{4}$	Aggregated Results	$KE_{q,\;ij}\;\left(x\right)$	$EN_{q,\;ij}\;\left(x\right)$	${ℇ}_{j}$	${ⱳ}_{j}$
${\mathrm{PSC}}_{1}$	GOO	VGO	EXG	VGO	[0.887, 0.253]	0.549576	0.450424	0.166067	0.278
${\mathrm{ASC}}_{2}$	FAR	MDG	MDG	GOO	[0.656, 0.461]	0.133121	0.866879	0.319610	0.226
${\mathrm{TGC}}_{3}$	GOO	VGO	VGO	VGO	[0.856, 0.260]	0.460183	0.539817	0.199025	0.266
${\mathrm{TPC}}_{4}$	MDG	POO	VGO	MDP	[0.538, 0.651]	0.144820	0.855180	0.315296	0.230

**Table 6 sensors-22-04123-t006:** Verbal evaluations of the protected system’s criteria and aggregated main criteria weights.

Sub-Criteria	$Ex_{1}$	$Ex_{2}$	$Ex_{3}$	$Ex_{4}$	Aggregated Results	$KE_{q,\;ij}\left(x\right)$	$EN_{q,\;ij}\left(x\right)$	${ℇ}_{j}$	${ⱳ}_{j}$
${\mathrm{HIC}}_{1\_1}$	MDP	FAR	VPO	EXG	[0.449, 0.748]	0.243794	0.756206	0.360331	0.319
${\mathrm{NIC}}_{1\_2}$	ELP	VGO	MDP	MDG	[0.445, 0.862]	0.484883	0.515117	0.245452	0.377
${\mathrm{HYC}}_{1\_3}$	VGO	GOO	GOO	POO	[0.667, 0.576]	0.172681	0.827319	0.394216	0.304

**Table 7 sensors-22-04123-t007:** Verbal evaluations of the audit source location’s criteria and aggregated main criteria weights.

Sub-Criteria	$Ex_{1}$	$Ex_{2}$	$Ex_{3}$	$Ex_{4}$	Aggregated Results	$KE_{q,\;ij}\left(x\right)$	$EN_{q,\;ij}\left(x\right)$	${ℇ}_{j}$	${ⱳ}_{j}$
${\mathrm{HLC}}_{2\_1}$	VPO	EXG	MDP	FAR	[0.510, 0.922]	0.684180	0.315820	0.151557	0.282
${\mathrm{NPC}}_{2\_2}$	ELP	VGO	MDP	MDG	[0.445, 0.862]	0.484883	0.515117	0.247197	0.251
${\mathrm{ALC}}_{2\_3}$	VGO	MDG	MDG	POO	[0.609, 0.588]	0.133588	0.866412	0.415779	0.196
${\mathrm{ISC}}_{2\_4}$	VGO	MDP	ELP	MDG	[0.361, 0.906]	0.613525	0.386475	0.185464	0.271

**Table 8 sensors-22-04123-t008:** Verbal evaluations of the targets’ criteria and aggregated main criteria weights.

Sub-Criteria	$Ex_{1}$	$Ex_{2}$	$Ex_{3}$	$Ex_{4}$	Aggregated Results	$KE_{q,\;ij}\left(x\right)$	$EN_{q,\;ij}\left(x\right)$	${ℇ}_{j}$	${ⱳ}_{j}$
${\mathrm{APC}}_{3\_1}$	FAR	MDG	MDG	GOO	[0.656, 0.461]	0.133121	0.866879	0.431377	0.284
${\mathrm{NEC}}_{3\_2}$	VGO	MDP	ELP	MDG	[0.361, 0.906]	0.613525	0.386475	0.192318	0.404
${\mathrm{HOC}}_{3\_3}$	MDP	FAR	VPO	EXG	[0.449, 0.748]	0.243794	0.756206	0.376304	0.312

**Table 9 sensors-22-04123-t009:** Verbal evaluations of the types’ criteria and aggregated main criteria weights.

Sub-Criteria	$Ex_{1}$	$Ex_{2}$	$Ex_{3}$	$Ex_{4}$	Aggregated Results	$KE_{q,\;ij}\left(x\right)$	$EN_{q,\;ij}\left(x\right)$	${ℇ}_{j}$	${ⱳ}_{j}$
${\mathrm{OSC}}_{4\_1}$	MDP	FAR	VPO	GOO	[0.427, 0.748]	0.241568	0.758432	0.356283	0.322
${\mathrm{CSC}}_{4\_2}$	MDG	POO	VGO	MDP	[0.538, 0.651]	0.144820	0.855180	0.401732	0.299
${\mathrm{FRC}}_{4\_3}$	ELP	VGO	MDP	MDG	[0.445, 0.862]	0.484883	0.515117	0.241983	0.379

**Table 10 sensors-22-04123-t010:** The global weights of the sub criteria for evaluating intrusion-detection systems.

**Main Criteria**	$PSC_{1}\;(0.278)$			$ASC_{2}\;(0.226)$			
Sub-criteria	${\mathrm{HIC}}_{1\_1}$	${\mathrm{NIC}}_{1\_2}$	${\mathrm{HYC}}_{1\_3}$	${\mathrm{HLC}}_{2\_1}$	${\mathrm{NPC}}_{2\_2}$	${\mathrm{ALC}}_{2\_3}$	${\mathrm{ISC}}_{2\_4}$
Local weights	0.319	0.377	0.304	0.282	0.251	0.196	0.271
Global weights	0.089	0.105	0.085	0.064	0.056	0.045	0.061
**Main Criteria**	${\mathrm{TGC}}_{3}$ **(0.266)**			${\mathrm{TPC}}_{4}$ **(0.230)**			
Sub-criteria	${\mathrm{APC}}_{3\_1}$	${\mathrm{NEC}}_{3\_2}$	${\mathrm{HOC}}_{3\_3}$	${\mathrm{OSC}}_{4\_1}$	${\mathrm{CSC}}_{4\_2}$	${\mathrm{FRC}}_{4\_3}$	
Local weights	0.284	0.404	0.312	0.322	0.299	0.379	
Global weights	0.075	0.107	0.083	0.074	0.069	0.087	

**Table 11 sensors-22-04123-t011:** Evaluations of the IDSs in terms of criteria.

IDS	$Ex_{s}$	$HIC_{1\_1}$	$NIC_{1\_2}$	$HYC_{1\_3}$	$HLC_{2\_1}$	$NPC_{2\_2}$	$ALC_{2\_3}$	$ISC_{2\_4}$	$APC_{3\_1}$	$NEC_{3\_2}$	$HOC_{3\_3}$	$OSC_{4\_1}$	$CSC_{4\_2}$	$FRC_{4\_3}$
${\mathrm{IDS}}_{1}$	${\mathrm{Ex}}_{1}$	VEH	VEH	VEH	VEH	MEH	MEH	HGH	HGH	HGH	VLO	VLO	VLO	HGH
	${\mathrm{Ex}}_{2}$	VEH	VEH	VEH	HGH	VEH	HGH	HGH	VEH	VEH	MEL	HGH	HGH	MED
	${\mathrm{Ex}}_{3}$	HGH	HGH	HGH	HGH	HGH	HGH	HGH	VEH	VEH	MED	MEL	MEL	MED
	${\mathrm{Ex}}_{4}$	HGH	HGH	HGH	VEH	EXH	HGH	HGH	EXH	VEH	HGH	VLO	HGH	MED
	${\mathrm{Ex}}_{s}$	${\mathrm{HIC}}_{1\_1}$	${\mathrm{NIC}}_{1\_2}$	${\mathrm{HYC}}_{1\_3}$	${\mathrm{HLC}}_{2\_1}$	${\mathrm{NPC}}_{2\_2}$	${\mathrm{ALC}}_{2\_3}$	${\mathrm{ISC}}_{2\_4}$	${\mathrm{APC}}_{3\_1}$	${\mathrm{NEC}}_{3\_2}$	${\mathrm{HOC}}_{3\_3}$	${\mathrm{OSC}}_{4\_1}$	${\mathrm{CSC}}_{4\_2}$	${\mathrm{FRC}}_{4\_3}$
${\mathrm{IDS}}_{2}$	${\mathrm{Ex}}_{1}$	HGH	LLO	EXO	VEH	LLO	MEL	HGH	VEH	VEH	VLO	VLO	VLO	VEH
	${\mathrm{Ex}}_{2}$	VEH	LLO	EXO	HGH	HGH	MEH	VEH	VEH	VEH	MEL	HGH	HGH	VLO
	${\mathrm{Ex}}_{3}$	HGH	LLO	EXO	HGH	MEL	MED	HGH	VEH	VEH	MED	MEL	MEL	MED
	${\mathrm{Ex}}_{4}$	HGH	LLO	EXO	EXH	VEH	MED	HGH	EXH	VEH	HGH	VLO	HGH	LLO
	${\mathrm{Ex}}_{s}$	${\mathrm{HIC}}_{1\_1}$	${\mathrm{NIC}}_{1\_2}$	${\mathrm{HYC}}_{1\_3}$	${\mathrm{HLC}}_{2\_1}$	${\mathrm{NPC}}_{2\_2}$	${\mathrm{ALC}}_{2\_3}$	${\mathrm{ISC}}_{2\_4}$	${\mathrm{APC}}_{3\_1}$	${\mathrm{NEC}}_{3\_2}$	${\mathrm{HOC}}_{3\_3}$	${\mathrm{OSC}}_{4\_1}$	${\mathrm{CSC}}_{4\_2}$	${\mathrm{FRC}}_{4\_3}$
${\mathrm{IDS}}_{3}$	${\mathrm{Ex}}_{1}$	HGH	VEH	VEH	VEH	HGH	MED	HGH	VEH	MEL	VLO	VLO	VLO	EXO
	${\mathrm{Ex}}_{2}$	VEH	VEH	VEH	HGH	EXH	VEH	VEH	VEH	MEL	MEL	HGH	HGH	EXO
	${\mathrm{Ex}}_{3}$	HGH	HGH	HGH	HGH	VEH	MEH	HGH	VEH	MEL	MED	MEL	MEL	EXO
	${\mathrm{Ex}}_{4}$	HGH	HGH	HGH	VEH	EXH	HGH	HGH	VEH	MEL	HGH	VLO	HGH	EXO
	${\mathrm{Ex}}_{s}$	${\mathrm{HIC}}_{1\_1}$	${\mathrm{NIC}}_{1\_2}$	${\mathrm{HYC}}_{1\_3}$	${\mathrm{HLC}}_{2\_1}$	${\mathrm{NPC}}_{2\_2}$	${\mathrm{ALC}}_{2\_3}$	${\mathrm{ISC}}_{2\_4}$	${\mathrm{APC}}_{3\_1}$	${\mathrm{NEC}}_{3\_2}$	${\mathrm{HOC}}_{3\_3}$	${\mathrm{OSC}}_{4\_1}$	${\mathrm{CSC}}_{4\_2}$	${\mathrm{FRC}}_{4\_3}$
${\mathrm{IDS}}_{4}$	${\mathrm{Ex}}_{1}$	VEH	VEH	VEH	VEH	LLO	MEL	HGH	HGH	MED	VLO	VLO	VLO	MEL
	${\mathrm{Ex}}_{2}$	VEH	VEH	VEH	HGH	HGH	MEH	VEH	HGH	MED	MEL	HGH	HGH	MEL
	${\mathrm{Ex}}_{3}$	HGH	HGH	HGH	HGH	MEL	MED	HGH	HGH	MED	MED	MEL	MEL	MEL
	${\mathrm{Ex}}_{4}$	HGH	HGH	HGH	EXH	VEH	MED	HGH	HGH	MED	HGH	VLO	HGH	MEL
	${\mathrm{Ex}}_{s}$	${\mathrm{HIC}}_{1\_1}$	${\mathrm{NIC}}_{1\_2}$	${\mathrm{HYC}}_{1\_3}$	${\mathrm{HLC}}_{2\_1}$	${\mathrm{NPC}}_{2\_2}$	${\mathrm{ALC}}_{2\_3}$	${\mathrm{ISC}}_{2\_4}$	${\mathrm{APC}}_{3\_1}$	${\mathrm{NEC}}_{3\_2}$	${\mathrm{HOC}}_{3\_3}$	${\mathrm{OSC}}_{4\_1}$	${\mathrm{CSC}}_{4\_2}$	${\mathrm{FRC}}_{4\_3}$
${\mathrm{IDS}}_{5}$	${\mathrm{Ex}}_{1}$	HGH	VEH	VLO	VEH	HGH	MED	HGH	VLO	VEH	VLO	VLO	VLO	HGH
	${\mathrm{Ex}}_{2}$	VEH	VEH	HGH	HGH	EXH	VEH	VEH	VLO	VEH	MEL	HGH	HGH	MED
	${\mathrm{Ex}}_{3}$	HGH	HGH	MEL	HGH	VEH	MEH	HGH	VLO	VEH	MED	MEL	MEL	MED
	${\mathrm{Ex}}_{4}$	HGH	HGH	VLO	VEH	EXH	HGH	HGH	VLO	VEH	HGH	VLO	HGH	MED
	${\mathrm{Ex}}_{s}$	${\mathrm{HIC}}_{1\_1}$	${\mathrm{NIC}}_{1\_2}$	${\mathrm{HYC}}_{1\_3}$	${\mathrm{HLC}}_{2\_1}$	${\mathrm{NPC}}_{2\_2}$	${\mathrm{ALC}}_{2\_3}$	${\mathrm{ISC}}_{2\_4}$	${\mathrm{APC}}_{3\_1}$	${\mathrm{NEC}}_{3\_2}$	${\mathrm{HOC}}_{3\_3}$	${\mathrm{OSC}}_{4\_1}$	${\mathrm{CSC}}_{4\_2}$	${\mathrm{FRC}}_{4\_3}$
${\mathrm{IDS}}_{6}$	${\mathrm{Ex}}_{1}$	HGH	VEH	VEH	VEH	HGH	MED	HGH	VEH	MEH	VLO	VLO	VLO	VLO
	${\mathrm{Ex}}_{2}$	VEH	VEH	VEH	HGH	EXH	VEH	VEH	VEH	MEH	MEL	HGH	HGH	VLO
	${\mathrm{Ex}}_{3}$	HGH	HGH	HGH	HGH	VEH	MEH	HGH	VEH	MEH	MED	MEL	MEL	VLO
	${\mathrm{Ex}}_{4}$	HGH	HGH	HGH	VEH	EXH	HGH	HGH	VEH	MEH	HGH	VLO	HGH	VLO

**Table 12 sensors-22-04123-t012:** The aggregated evaluations matrix of the IDSs.

**IDS**	$HIC_{1\_1}$	$NIC_{1\_2}$	$HYC_{1\_3}$	$HLC_{2\_1}$	$NPC_{2\_2}$	$ALC_{2\_3}$	$ISC_{2\_4}$
${\mathrm{IDS}}_{1}$	[0.823, 0.295]	[0.823, 0.295]	[0.823, 0.295]	[0.823, 0.295]	[0.817, 0.342]	[0.747, 0.365]	[0.770, 0.330]
${\mathrm{IDS}}_{2}$	[0.801, 0.318]	[0.330, 0.770]	[0.110, 0.990]	[0.832, 0.301]	[0.564, 0.630]	[0.555, 0.562]	[0.801, 0.311]
${\mathrm{IDS}}_{3}$	[0.801, 0.318]	[0.823, 0.295]	[0.823, 0.295]	[0.832, 0.301]	[0.909, 0.248]	[0.715, 0.438]	[0.801, 0.311]
${\mathrm{IDS}}_{4}$	[0.823, 0.295]	[0.823, 0.295]	[0.823, 0.295]	[0.832, 0.301]	[0.817, 0.342]	[0.555, 0.562]	[0.801, 0.311]
${\mathrm{IDS}}_{5}$	[0.801, 0.318]	[0.823, 0.295]	[0.394, 0.780]	[0.812, 0.303]	[0.909, 0.284]	[0.715, 0.438]	[0.801, 0.318]
${\mathrm{IDS}}_{6}$	[0.801, 0.318]	[0.823, 0.295]	[0.823, 0.295]	[0.812, 0.303]	[0.909, 0.284]	[0.715, 0.438]	[0.801, 0.318]
**IDS**	${\mathrm{APC}}_{3\_1}$	${\mathrm{NEC}}_{3\_2}$	${\mathrm{HOC}}_{3\_3}$	${\mathrm{OSC}}_{4\_1}$	${\mathrm{CSC}}_{4\_2}$	${\mathrm{FRC}}_{4\_3}$	
${\mathrm{IDS}}_{1}$	[0.877, 0.259]	[0.857, 0.260]	[0.458, 0.713]	[0.394, 0.780]	[0.507, 0.704]	[0.588, 0.528]	
${\mathrm{IDS}}_{2}$	[0.901, 0.211]	[0.880, 0.220]	[0.458, 0.715]	[0.441, 0.661]	[0.507, 0.706]	[0.414, 0.765]	
${\mathrm{IDS}}_{3}$	[0.880, 0.220]	[0.440, 0.660]	[0.458, 0.714]	[0.394, 0.780]	[0.507, 0.705]	[0.110, 0.990]	
${\mathrm{IDS}}_{4}$	[0.770, 0.330]	[0.550, 0.550]	[0.458, 0.714]	[0.394, 0.780]	[0.507, 0.705]	[0.440, 0.432]	
${\mathrm{IDS}}_{5}$	[0.220, 0.880]	[0.880, 0.220]	[0.458, 0.714]	[0.394, 0.780]	[0.507, 0.705]	[0.588, 0.528]	
${\mathrm{IDS}}_{6}$	[0.880, 0.220]	[0.660, 0.440]	[0.458, 0.714]	[0.394, 0.780]	[0.507, 0.705]	[0.220, 0.880]	

**Table 13 sensors-22-04123-t013:** The normalized evaluation matrix of the IDSs.

**IDS**	$HIC_{1\_1}$	$NIC_{1\_2}$	$HYC_{1\_3}$	$HLC_{2\_1}$	$NPC_{2\_2}$	$ALC_{2\_3}$	$ISC_{2\_4}$
${\mathrm{IDS}}_{1}$	[0.823, 0.295]	[0.823, 0.295]	[0.823, 0.295]	[0.823, 0.295]	[0.817, 0.342]	[0.747, 0.365]	[0.770, 0.330]
${\mathrm{IDS}}_{2}$	[0.801, 0.318]	[0.330, 0.770]	[0.110, 0.990]	[0.832, 0.301]	[0.564, 0.630]	[0.555, 0.562]	[0.801, 0.311]
${\mathrm{IDS}}_{3}$	[0.801, 0.318]	[0.823, 0.295]	[0.823, 0.295]	[0.832, 0.301]	[0.909, 0.248]	[0.715, 0.438]	[0.801, 0.311]
${\mathrm{IDS}}_{4}$	[0.823, 0.295]	[0.823, 0.295]	[0.823, 0.295]	[0.832, 0.301]	[0.817, 0.342]	[0.555, 0.562]	[0.801, 0.311]
${\mathrm{IDS}}_{5}$	[0.801, 0.318]	[0.823, 0.295]	[0.394, 0.780]	[0.812, 0.303]	[0.909, 0.284]	[0.715, 0.438]	[0.801, 0.318]
${\mathrm{IDS}}_{6}$	[0.801, 0.318]	[0.823, 0.295]	[0.823, 0.295]	[0.812, 0.303]	[0.909, 0.284]	[0.715, 0.438]	[0.801, 0.318]
**IDS**	${\mathrm{APC}}_{3\_1}$	${\mathrm{NEC}}_{3\_2}$	${\mathrm{HOC}}_{3\_3}$	${\mathrm{OSC}}_{4\_1}$	${\mathrm{CSC}}_{4\_2}$	${\mathrm{FRC}}_{4\_3}$	
${\mathrm{IDS}}_{1}$	[0.877, 0.259]	[0.857, 0.260]	[0.458, 0.713]	[0.394, 0.780]	[0.507, 0.704]	[0.588, 0.528]	
${\mathrm{IDS}}_{2}$	[0.901, 0.211]	[0.880, 0.220]	[0.458, 0.715]	[0.441, 0.661]	[0.507, 0.706]	[0.414, 0.765]	
${\mathrm{IDS}}_{3}$	[0.880, 0.220]	[0.440, 0.660]	[0.458, 0.714]	[0.394, 0.780]	[0.507, 0.705]	[0.110, 0.990]	
${\mathrm{IDS}}_{4}$	[0.770, 0.330]	[0.550, 0.550]	[0.458, 0.714]	[0.394, 0.780]	[0.507, 0.705]	[0.440, 0.432]	
${\mathrm{IDS}}_{5}$	[0.220, 0.880]	[0.880, 0.220]	[0.458, 0.714]	[0.394, 0.780]	[0.507, 0.705]	[0.588, 0.528]	
${\mathrm{IDS}}_{6}$	[0.880, 0.220]	[0.660, 0.440]	[0.458, 0.714]	[0.394, 0.780]	[0.507, 0.705]	[0.220, 0.880]	

**Table 14 sensors-22-04123-t014:** The ${ἁ}_{j}$ and ${ẞ}_{j}$ values of the IDSs.

IDSs	Wq-ROFHA Operator		Wq-ROFHGM Operator	
	ἁ	ẞ	ἁ	ẞ
${\mathrm{IDS}}_{1}$	0.784	0.382	0.139	0.567
${\mathrm{IDS}}_{2}$	0.741	0.489	0.102	0.789
${\mathrm{IDS}}_{3}$	0.767	0.438	0.113	0.777
${\mathrm{IDS}}_{4}$	0.741	0.421	0.127	0.578
${\mathrm{IDS}}_{5}$	0.760	0.483	0.119	0.671
${\mathrm{IDS}}_{6}$	0.771	0.419	0.125	0.651

**Table 15 sensors-22-04123-t015:** The score values of the IDSs for ${ἁ}_{j}$ and ${ẞ}_{j}$.

IDS	$${ἁ}_{j}$$	$${ẞ}_{j}$$
${\mathrm{IDS}}_{1}$	0.701	0.286
${\mathrm{IDS}}_{2}$	0.626	0.156
${\mathrm{IDS}}_{3}$	0.665	0.168
${\mathrm{IDS}}_{4}$	0.660	0.275
${\mathrm{IDS}}_{5}$	0.639	0.224
${\mathrm{IDS}}_{6}$	0.676	0.237

**Table 16 sensors-22-04123-t016:** The proportional importance and the final ranking of the IDSs.

IDS	${Ɣ}_{ja}$	${Ɣ}_{jb}$	${Ɣ}_{jc}$	${Ɣ}_{j}$	Rank
${\mathrm{IDS}}_{1}$	0.186	2.952	1.000	2.398	1
${\mathrm{IDS}}_{2}$	0.147	2.000	0.792	1.595	6
${\mathrm{IDS}}_{3}$	0.157	2.136	0.843	1.701	5
${\mathrm{IDS}}_{4}$	0.176	2.814	0.947	2.089	2
${\mathrm{IDS}}_{5}$	0.162	2.455	0.874	1.867	4
${\mathrm{IDS}}_{6}$	0.172	2.597	0.925	1.976	3

**Table 17 sensors-22-04123-t017:** Comparative analysis with other approach for ranking the IDSs.

Approaches	$IDS_{1}$	$IDS_{2}$	$IDS_{3}$	$IDS_{4}$	$IDS_{5}$	$IDS_{6}$
q-ROF Entropy-CoCoSo	2.398	1.595	1.701	2.089	1.867	1.976
Ranking	1	6	5	2	4	3
Fuzzy AHP-TOPSIS	0.927	0.287	0.582	0.675	0.497	0.723
Ranking	1	6	4	3	5	2

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
