# Peer review of "An Optimization Model for Appraising Intrusion-Detection Systems for Network Security Communications: Applications, Challenges, and Solutions"

_sensors, 2022, doi:10.3390/s22114123_

Round 1
Reviewer 1 Report
In general, the article is of high quality, a few suggestions should be taken into account.
- In sub-section 3.1.3, it necessary to note that q is integer and >=1.
- In Eq. (12), possibly powers q are omitted.
- Tables 1 and 2 conatin the same values of q-ROFNs; it would be better to combine them and make one table with 6 columns.
- The choice of very high values of parameter q in Section 4.5 looks unnecessary; from Fig. 2 it is seen that q>>1 corresponds to curves, almost coinciding with unity square; in connection with this, the results for q>10 may contain high degree of computation errors.
Author Response
Revisions in response to the comments from Reviewer #1
Comments to the Author.
In general, the article is of high quality, a few suggestions should be taken into account:
Response: We would like to first thank you for your valuable comments and appreciate the time that you spent for reviewing our work. We also commend your vigilance in finding the oversights. We have addressed all your comments as follows and hope you find them satisfactory:
Comment 1: In sub-section 3.1.3, it necessary to note that q is integer and >=1.
Response: Many thanks for this comment. We checked this note based on your comment and hope you find them satisfactory.
Comment 2: In Eq. (12), possibly powers q are omitted.
Response: Many thanks for this comment. This equation is updated in the revised manuscript based on your comment and hope you find them satisfactory. Please see page 7, line 302.
Eq. 12 is copied below for you to review.
0 ≤ ≤ 1, for Ɐ (12)
Comment 3: Tables 1 and 2 contain the same values of q-ROFNs; it would be better to combine them and make one table with 6 columns.
Response: Many thanks for this comment. This part was updated in the revised manuscript based on your comment and hope you find them satisfactory.
The relevant table is copied below for you to review.
Table 1. Verbal variables and their corresponding q-ROFNs for weighting criteria and ranking alternatives.
|
Verbal variables for criteria |
Abbreviations for criteria |
Verbal variables for alternatives |
Abbreviations for alternatives |
q-ROFNs |
|
|
μ |
ʋ |
||||
|
Extremely poor |
ELP |
Extremely low |
EXO |
0.11 |
0.99 |
|
Very poor |
VPO |
Very low |
VLO |
0.22 |
0.88 |
|
Poor |
POO |
Low |
LLO |
0.33 |
0.77 |
|
Medium poor |
MDP |
Medium low |
MEL |
0.44 |
0.66 |
|
Fair |
FAR |
Medium |
MED |
0.55 |
0.55 |
|
Medium good |
MDG |
Medium high |
MEH |
0.66 |
0.44 |
|
Good |
GOO |
High |
HGH |
0.77 |
0.33 |
|
Very good |
VGO |
Very high |
VEH |
0.88 |
0.22 |
|
Extremely good |
EXG |
Extremely high |
EXH |
0.99 |
0.11 |
Comment 4: The choice of very high values of parameter q in Section 4.5 looks unnecessary; from Fig. 2 it is seen that q>>1 corresponds to curves, almost coinciding with unity square; in connection with this, the results for q>10 may contain a high degree of computation errors.
Response: Many thanks for this comment. We checked this note based on your comment and hope you find them satisfactory. Multiple values of the parameter q were used to show the changes in the results. The value of q > 10 shows the same direction of data change for the previous values of the parameter q and does not indicate any errors in the calculation.
At the end, the authors wish to thank the reviewer for the detailed review comments. We hope you will find the paper acceptable this time.

Reviewer 2 Report
A novel hybrid MCDM approach is proposed for the evaluation of the IDS.
To improve the quality of the paper, a few suggestions are as follows.
1. The literature review presented here is highly insufficient and generalized. Please improve it using recent papers.
2. Eqn. 2,3, 26,27 are not clear. Please elaborate.
3. Few variables are not defined. Please correct it.
4. The picture quality should be improved.
5. Few short forms have been used without giving full forms. Please cross-check throughout the paper properly.
6. To improve the introduction and reference sections, you should follow quality papers.
[1] S. Cheng, S. Jianfu, M. Alrasheedi, P. Saeidi, A. R. Mishra, and P. Rani, "A new extended VIKOR approach using q-rung orthopair fuzzy sets for sustainable enterprise risk management assessment in manufacturing small and medium-sized enterprises," International Journal of Fuzzy Systems, vol. 23, no. 5, pp. 1347-1369, 2021.
[2] H. Li, S. Yin, and Y. Yang, "Some preference relations based on q‐rung orthopair fuzzy sets," International Journal of Intelligent Systems, vol. 34, no. 11, pp. 2920-2936, 2019.
[3] X. Peng, J. Dai, and H. Garg, "Exponential operation and aggregation operator for q‐rung orthopair fuzzy set and their decision‐making method with a new score function," International Journal of Intelligent Systems, vol. 33, no. 11, pp. 2255-2282, 2018.
Please go through those papers, and include and improve your literature review portion of the paper.
8. Elaborate discussions of results. Try to point out each waveform using proper justification.
9. Rewrite the conclusion section in the summarized form.
- Some sentences should be rewritten.
Author Response
Revisions in response to the comments from Reviewer #2
Comment 1: - The contributions of the paper can be more specific and quantified, e.g. how much is the improvement of the performance by %? What are the originalities? What are the main differences between this papers with other papers, instead of stating the investigation area?
Response: Many thanks for this comment. More explanations have been added in different places. Also, we revised the introduction section to highlight the contribution of the proposed work (please see pages 4 and 5). We have explained the research motivation, justification of proposing methods for the evaluation of the IDS, the applicability of the model, and also a brief research methodology is highlighted in this revised introduction section. Results and discussions have been rewritten to summarize the findings/significance of the work.
Comment 2: Discuss literature reviews based on their performance metrics along with the boons and limitations of each work surveyed.
Response: Many thanks for this comment. We explained this point with more details in this version.
Part of the relevant text is copied below for you to review.
This section provides basic knowledge about IDSs to enable a deeper understanding of this topic. Also, some basic information related to MCDM has been presented. Afterward, some studies related to q-ROF theory have been introduced. Alyami et al. presented a study based on a hybrid MCDM approach consisting of the fuzzy analytical hierarchy process (AHP) and the fuzzy technique for order performance by similarity to ideal solution (TOPSIS) to evaluate the effectiveness of IDSs [16]. In their study, they used four main criteria and thirteen sub-aspects to evaluate five IDSs. Their results indicate that Suricata is the most effective IDS. Also, their results indicate that most of the IDSs that were evaluated in the study are effective and close in their results. B. Abushark et al. developed a study to evaluate the optimization of machine learning-based IDSs using a hybrid MCDM approach which comprises the AHP and TOPSS in a fuzzy environment [24]. Their findings aim to identify attributes related to cyber security, allowing the design of more effective and efficient IDSs. Al-Harbi et al. presented a study for an optimal evaluation of machine learning-based IDSs using a hybrid MCDM model that includes AHP and TOPSIS under hesitant fuzzy conditions [8]. Their findings aim to identify features related to cyber security, allowing the design of more effective and IDSs. Almotiri presented an evaluation system to detect malicious traffic based on system performance [25]. They adopted an MCDM approach consisting of AHP-TOPSIS methods to rank the impact of alternatives according to their overall performance. Their study aims to be a reference for practitioners working in the field of evaluating and selecting the most effective traffic detection approach.
Afterward, some studies related to MCDM approaches and their applications in different fields are presented. Sharma and Kaul presented a study to deal with network performance delays and disruptions due to cluster-based communications that place a significant burden on the cluster head (CH) [26].
Comment 3: Novelty of the work is also a great question? Authors need to justify it.
Response: Many thanks for this comment. This part was updated in the revised manuscript based on your comment and hope you find them satisfactory. Also, more explanations have been added in different places. Please see pages 4-5.
Part of the relevant text is copied below for you to review.
In this study, a set of criteria were adopted to evaluate IDS according to previous studies and expert opinions. The criteria that have been adopted are divided into four basic criteria are protected system, audit source location, alerts, and types, and each main criterion includes several sub-aspects. The set of sub-aspects namely: HIDS, NIDS, hybrids, host log files, network packets, application log files, IDS sensor application, network, host, open-source, closed source, and freeware. In order to solve such complex problems related to the evaluation of IDSs, multi-criteria decision-making (MCDM) has been proven to be one of the best tools for the effective evaluation of IDS [16]. MCDM is popular in complex problems because it enables the decision-maker to take care of all the available criteria and take an appropriate decision as per the priority [17]. Since the ideal choice is governed by multiple criteria, a good decision-maker, in certain situations, may look for criteria of high impact on which to focus.
Consequently, the assessment of IDS under multiple criteria and a pluralistic viewpoint, the assessment process is tainted by ambiguity and uncertainty, which is difficult to deal with in real numbers. Hence, the q-rung orthopair fuzzy sets (q-ROFSs) theory has been applied to deal with such complex problems [18]. The q-ROFS proved to be effective in solving ambiguous and uncertain problems as it came as a generalization of the intuitionistic fuzzy sets (IFSs) [19], and Pythagorean fuzzy sets (PFSs) theories [20].
Finally, to deal with the problem of evaluating the effectiveness of IDSs, a hybrid approach consisting of two multi-criteria decision-making methods, the entropy method [21], and the combined compromised solution (CoCoSo) method [22], was adopted. The proposed hybrid approach is presented under the q-rung orthopair fuzzy environment and by utilizing the q-rung orthopair fuzzy numbers (q-ROFNs) numbers. Firstly, the entropy method was adopted to evaluate the main and sub-aspects and to determine the final weights. Secondly, the CoCoSo method was applied to evaluate the available alternatives and determine the best alternative.
Comment 4: What is the computation time for the algorithm?Provide running time for the proposed method? Provide the comparison of computation time between the proposed method and other works.
Response: Many thanks for this comment. The study proposes hybrid multi-criteria decision-making approach for evaluating IDS, not an algorithm.
Comment 5: If the proposed method is highly accurate, can you please provide some failure cases for the method? And discuss the underlying cause of these failure cases.
Response: Many thanks for this comment.
Comment 6: It is interesting to see the accuracy of the proposed method in different training sets with different sizes. Please provide the comparison with other works on this issue.
Response: Many thanks for this comment. According to our proposed approach to multi-criteria decision-making, we made a comparison with other works. Please see page 19, line 631-646.
The relevant text is copied below for you to review.
Comparative analysis
In this sub-section, a comparative analysis has been demonstrated to test and verify the effectiveness of the developed approach q-ROF Entropy-CoCoSo. Consequently, the assessment results have been compared with Alyami et al.,'s [16] fuzzy AHP-TOPSIS approach. In this regard, the same weights of the main criteria and sub-aspects obtained by applying the proposed approach were used as shown in Table 10. Accordingly, the results of ranking the alternatives used in the study using the two approaches are presented in Table 17 and in Fig 7. The results of the comparison show that is the best alternative according to the results of the two approaches. Whereas is the least alternative in the order. According to the results, it can be seen that there are some changes in the order of some alternatives such as , , , and . The presence of some differences in the order of the alternatives can be explained by the difference in the mathematical basis for each approach. Finally, the results of the comparative analysis and the reliability of the proposed approach can be verified by the experts.
Table 17. Comparative analysis with other approach for ranking IDSs.
|
Approaches |
|
|
|
|
|
|
|
q-ROF Entropy-CoCoSo |
2.398 |
1.595 |
1.701 |
2.089 |
1.867 |
1.976 |
|
Ranking |
1 |
6 |
5 |
2 |
4 |
3 |
|
Fuzzy AHP-TOPSIS |
0.927 |
0.287 |
0.582 |
0.675 |
0.497 |
0.723 |
|
Ranking |
1 |
6 |
4 |
3 |
5 |
2 |
Comment 7: There are many algorithm parameters in the proposed method. What's the influence of these parameters?
Response: Many thanks for this comment. We explained this part in the sensitivity analysis section based on your comment and hope you find them satisfactory. Please see pages 8 and 20.
Part of the relevant text is copied below for you to review.
We have been conducted a sensitivity analysis from the three perspectives of changes in the parameter q, parameter γ and parameter changes. Sensitivity analysis was conducted on the results obtained to confirm their reliability and stability and to examine the change that occurred to them as a result of the change in some inputs and parameters. In decision-making approaches, some parameters are defined subjectively based on the perception of the problem by decision-makers and the extent of the risks in the environment. Consequently, these parameters change according to the circumstances in which the decision-making system is being modeled. In our proposed Entropy-CoCoSo q-ROF approach, three parameters q, γ, and are defined which are determined based on the personal preferences of the experts. Accordingly, several changes were made to these parameters to show their decisive influence on the final IDSs ranking results. These changes were divided into four scenarios. The first scenario, refers to the change in the values of parameter q. Also, the second scenario indicates the change in the values of parameter γ. afterwards, the third scenario refers to the change in the values of parameters q and γ. In addition, the fourth scenario refers to the change in the values of parameter Ѱ.
Comment 8: Concluding remarks are missing from the related works section. What conclusion the authors have reached after the rigorous study of mentioned papers. These should be reflected in the manuscript.
Response: Many thanks for this comment. This part was updated in the revised manuscript based on your comment and hope you find them satisfactory.
Comment 9: Motivating the design choices made, providing details about the alternatives.
Response: Many thanks for this comment. This part was updated in the revised manuscript based on your comment and hope you find them satisfactory.
The relevant text is copied below for you to review.
4.1. Description of intrusion detection systems
In this subsection, a brief descriptions of intrusion detection systems are considered as follows. Also, Fig 4 demonstrates the general structure of the network and IDS.
- Suricata ( ): Suricata was developed by the Open Information Security Foundation in 2010. Suricata is the main alternative to snort because the design of Suricata is very close to that of snort [35]. Suricata has an advantage over Snort, which is that it collects data at the application layer. Suricata consists of so-called threads, thread units, and queues. Suricata is a multi-threaded program, so there will be multiple threads running at the same time [35]. Thread units are divided according to functions, for example, one unit is used to analyze data packets, and the other unit is used to discover data packets. Each data packet can be processed by several different threads, and the queue is used to transfer the data packet from one thread to another. At the same time, a thread can contain several thread units, but only one unit runs at a given time.
- Zeek ( ): Zeek (previously known as Bro until 2019) is a network intrusion detection system that is compatible with Linux, Unix, and Mac OS [36]. Zeek uses network-based intrusion detection methods by tracking the network and searching for malicious activities. The Zeek intrusion detection function is realized in two stages, traffic logging, and analysis. As with Suricata, Zeek has a significant advantage over Snort in that its analysis runs at the application layer, resulting in a broader analysis of network protocol activity.
- Security onion ( ): Security Onion is a Linux-based IDS that is a mixture of several IDS that are both HIDS and NIDS solutions [16]. Although Security Onion is classified as NIDS, it includes HIDS functionality as well. It monitors log and configuration files for suspicious activity and checks those files for any unexpected changes. One of the downsides to Security Onion's comprehensive network monitoring system is its complexity. Thus, the Security Onion analysis engine is where things get complicated because there are so many different tools with different operating procedures that most of them may end up being overlooked.
- Snort ( ): Snort is a Linux-based lightweight cross-platform network intrusion detection system that can be used to monitor TCP/IP networks [37]. Snort is easy to deploy and can be configured to monitor network traffic for intrusion attempts, log intrusion behavior, and perform specific actions when intrusion attempts are detected. It is one of the most widely deployed IDS tools and can also be used as an intrusion-prevention systems. Snort can be traced back to 1998, and there are still no signs of disappearing. There are some active communities that offer good help and support. The high level of personalization that Snort provides makes it a good choice for many different organizations.
- Wazuh ( ): Wazuh is an IDS used to detect security and monitor compliance with security rules. Wazuh is an open-source intrusion detection system project. It was developed as a fork part of OSSEC HIDS and was later integrated with Elastic Stack and Opens CAP. It relies on a cross-platform approach that redirects system data such as log messages, file tables, and detected anomalies to a central manager, where it is further analyzed and processed, resulting in security alerts. It monitors the file system and identifies changes in content, permissions, ownership, and file properties that need to be monitored. It monitors configuration files to ensure that they comply with security policies, standards, or hardening guides.
- OSSEC ( ): OSSEC is an open source IDS has been developed by Daniel B. Cid, who had been sold the system to Trend Micro in 2008 [37]. Its detection methods are based on checking log files, making it a host-based IDS. OSSEC works on Unix, Linux, Mac OS, and Windows. There is no front end for this tool, but you can connect with Kibana or Graylog. OSSEC disclosure rules are called 'Policies'. You can write your own policies or get packages of them for free from the user community. It is also possible to define actions that should be performed automatically when specific warnings appear.
Comment 10: Providing a clear yet mathematically rigorous description of the tools used
Response: Many thanks for this comment. This part was updated in the revised manuscript based on your comment and hope you find them satisfactory.
Comment 11: The quantitative and qualitative approach in the paper conclusions shall be significantly strengthen
Response: Many thanks for this comment. This part was updated in the revised manuscript based on your comment and hope you find them satisfactory.
In the end, the authors wish to thank the reviewer for the detailed review comments. We hope you will find the paper acceptable this time.

Reviewer 3 Report
-
- The contributions of the paper can be more specific and quantified, e.g. how much is the improvement of the performance by %? What are the originalities? What are the main differences between this papers with other papers, instead of stating the investigation area?
- Discuss literature reviews based on their performance metrics along with the boons and limitations of each work surveyed.
- Novelty of the work is also a great question? Authors need to justify it
- What is the computation time for the algorithm?Provide running time for the proposed method? Provide the comparison of computation time between the proposed method and other works.
- If the proposed method is highly accurate, can you please provide some failure cases for the method? And discuss the underlying cause of these failure cases.
- It is interesting to see the accuracy of the proposed method in different training sets with different sizes. Please provide the comparison with other works on this issue.
- There are many algorithm parameters in the proposed method. What's the influence of these parameters?
- Concluding remarks are missing from the related works section. What conclusion the authors have reached after the rigorous study of mentioned papers. These should be reflected in the manuscript.
- motivating the design choices made, providing details about the alternatives
- providing a clear yet mathematically rigorous description of the tools used - The quantitative and qualitative approach in the paper conclusions shall be significantly strengthen
Author Response
Revisions in response to the comments from Reviewer #3
Comments to the Author.
The paper provides an optimization model for appraising intrusion detection systems using a multi-criteria decision-making approach. The paper is interesting and generally well written. However, some minor issues need to be solved.
Response: we would like to first thank you for your valuable comments and appreciate the time that you spent for reviewing our work. We also admire your vigilance in finding the oversights. We have addressed all your comments as follows and hope you find them satisfactory:
Comment 1: Introduction section can be drastically reduced. It is too long and it contains some general information about Internet, networks, or IDSs that are not necessarily needed to describe what authors really want to present.
Response: Many thanks for this comment. Introduction section was updated in the revised manuscript based on your comment and hope you find them satisfactory.
Part of the relevant text is copied below for you to review.
The continuous development of computer systems has led to the increasing dependence of companies, organizations, and people on computer networks in performing their functions and offering their services in modern ways [1]. However, at the same time, it has become vulnerable to penetration by attackers with the aim of making illegal gains by exploiting some security vulnerabilities, which led to an increase in interest in issues of protection and security of these systems. Today there are many methods used within this field, and there are many intrusion detection systems (IDSs) available [2]. IDSs are a necessity for the stability of the organization's normal system performance. In this regard, traditional intrusion detection techniques are highly unrewarding and ineffective due to the multiplicity of attack methods and their different forms. Among the traditional methods used previously are obfuscation, transformation, and polymorphism techniques, which lead to malware resistance [3]. Despite the prominent role she plays, she still has some shortcomings. Therefore, there was a need to continue conducting research on intrusion detection systems in order to reach an optimal structure that achieves a high protection rate.
Comment 2: line 52: in two places "she" needs to be replaced by "it".
Response: Many thanks for this comment. Introduction section was updated in the revised manuscript based on your comment and hope you find them satisfactory. Please see page 2, line 52.
The relevant text is copied below for you to review.
Despite the prominent role it plays, it still has some shortcomings. Therefore, there was a need to continue conducting research on intrusion detection systems in order to reach an optimal structure that achieves a high protection rate.
Comment 3: Back matter sections (as requested by the journal's Instructions for Authors - e.g. "Funding", "Acknowledgments", "Author Contributions", etc.) need to be included at the end of the paper.
Response: Many thanks for this comment. This part was updated in the revised manuscript based on your comment and hope you find them satisfactory.
In the end, the authors wish to thank the reviewer for the detailed review comments. We hope you will find the paper acceptable this time.

Reviewer 4 Report
The paper provides an optimization model for appraising intrusion detection system using a multi-criteria decision-making approach. The paper is interesting and generally well written. However, some minor issues need to be solved:
- Introduction section can be drastically reduced. It is too long and it contains some general information about Internet, networks, or IDSs that are not necessarily needed to describe what authors really want to present.
- line 52: in two places "she" needs to be replaced by "it";
- Back matter sections (as requested by journal's Instructions for Authors - e.g. "Funding", "Acknowledgments", "Author Contributions", etc.) need to be included at the end of the paper.
Author Response
Revisions in response to the comments from Reviewer #4
Comments to the Author.
A novel hybrid MCDM approach is proposed for the evaluation of the IDS. To improve the quality of the paper, a few suggestions are as follows:
Response: we would like to first thank you for your valuable comments and appreciate the time that you spent for reviewing our work. We also admire your vigilance in finding the oversights. We have addressed all your comments as follows and hope you find them satisfactory:
Comment 1: The literature review presented here is highly insufficient and generalized. Please improve it using recent papers.
Response: Many thanks for this comment. Literature review section was updated in the revised manuscript based on your comment and hope you find them satisfactory.
The relevant text is copied below for you to review.
This section provides basic knowledge about IDSs to enable a deeper understanding of this topic. Also, some basic information related to MCDM has been presented. Afterward, some studies related to q-ROF theory have been introduced. Alyami et al. presented a study based on a hybrid MCDM approach consisting of the fuzzy analytical hierarchy process (AHP) and the fuzzy technique for order performance by similarity to ideal solution (TOPSIS) to evaluate the effectiveness of IDSs [16]. In their study, they used four main criteria and thirteen sub-aspects to evaluate five IDSs. Their results indicate that Suricata is the most effective IDS. Also, their results indicate that most of the IDSs that were evaluated in the study are effective and close in their results. B. Abushark et al. developed a study to evaluate the optimization of machine learning-based IDSs using a hybrid MCDM approach which comprises the AHP and TOPSS in a fuzzy environment [24]. Their findings aim to identify attributes related to cyber security, allowing the design of more effective and efficient IDSs. Al-Harbi et al. presented a study for an optimal evaluation of machine learning-based IDSs using a hybrid MCDM model that includes AHP and TOPSIS under hesitant fuzzy conditions [8]. Their findings aim to identify features related to cyber security, allowing the design of more effective and IDSs. Almotiri presented an evaluation system to detect malicious traffic based on system performance [25]. They adopted an MCDM approach consisting of AHP-TOPSIS methods to rank the impact of alternatives according to their overall performance. Their study aims to be a reference for practitioners working in the field of evaluating and selecting the most effective traffic detection approach.
Afterward, some studies related to MCDM approaches and their applications in different fields are presented. Sharma and Kaul presented a study to deal with network performance delays and disruptions due to cluster-based communications that place a significant burden on the cluster head (CH) [26]. They used an MCDM approach including two AHP-TOPSIS methods to reduce the overburden on a single CH through a multi-(CH) scheme. Ogundoyin and Kamil presented a study to address security and privacy issues where fog servers can be used to process private and respond to time-sensitive information [27]. They applied the fuzzy AHP MCDM approach to determine and prioritize confidence parameters in fog computing. Their results indicate that quality of service is the best priority parameter that a service requester can use to evaluate the trusted standard of a service provider. Kumar et al. introduced a study to evaluate the impact of various malware analysis methods on the perspective of web applications [28]. They applied an MCDM approach that comprises AHP and TOPSIS methods under fuzzy environment. Their results indicate that reverse engineering is the most effective method for analyzing complex malware.
There are many theories dealing with uncertainty, including the q-ROFS theory. Thus, we present some related studies as follows. Duane et al. introduced a study to deal with risks in information sharing and software piracy, which poses a threat to any system [29]. They applied the q-rung orthopair double hierarchy linguistic term set (q-RODHLTS) in the MCDM process. To prove the validity of their results, they applied the proposed approach to many information security systems. Panetikul et al. presented a study to analyze computer security threat analysis and control under a q-ROF environment [30]. They applied an approach based on the combination of the Heronian mean (HM) operator with complex q-ROFS is to initiate the complex q-rung orthopair fuzzy HM (Cq-ROFHM) operator. To prove the reliability and efficiency of the techniques used, some illustrative examples are introduced. Cheng et al. presented a study for evaluating sustainable enterprise risk management in manufacturing small and medium-sized enterprises [31]. They adopted a new extended Vlse Kriterijumska Optimizacija Kompromisno Resenje (VIKOR) approach using q-ROFSs. Peng et al. introduced a study for presenting a new score function of q-ROFN for solving the failure issues when comparing two q-ROFNs [32].
Finally, given the importance of IDSs and the importance of implementing them in a manner appropriate to their specific situation, choosing the most effective and appropriate is a great challenge. Hence, there is an urgent and great need to evaluate IDSs. In this regard, a set of main and sub-criteria affecting the selection of the most effective IDS has been identified. Also, a set of alternatives were identified to be evaluated according to these criteria, using an MCDM approach and under q-ROF environment.
Comment 2: Eqn. 2, 3, 26, 27 are not clear. Please elaborate.
Response: Many thanks for this comment. This part was updated in the revised manuscript based on your comment and hope you find them satisfactory. Please see pages 6 and 12, lines 254-260, and 330-337.
The relevant text is copied below for you to review.
= (1)
Where the function : Ҳ → [0, 1] describes the grade of membership of an element to the sets and : Ҳ → [0, 1] describes the grade of non-membership of an element to the sets , with the condition that:
0 ≤ ≤ 1, for Ɐ (2)
The grade of hesitancy is computed as follows:
= 1- (3)
Definition 10. Darko and Liang presented an operator named the weighted q-rung orthopair fuzzy Hamacher geometric mean (Wq-ROFHGM) [33] as in Eqs. (26), (27). Let = (i = 1, 2... n) be set of q-ROFNs and Ⱳ= be weight vector of with = 1 and [0, 1].
Wq-ROFHGM = ... = (26)
Wq- ROFHGM = (27)
Where > 0, q ≥ 0.
Comment 3: Few variables are not defined. Please correct it.
Response: Many thanks for this comment. This part was updated in the revised manuscript based on your comment and hope you find them satisfactory.
Comment 4: The picture quality should be improved.
Response: Many thanks for this comment. All figures quality were improved in the revised manuscript based on your comment and hope you find them satisfactory.
Comment 5: Few short forms have been used without giving full forms. Please cross-check throughout the paper properly.
Response: Many thanks for this comment. All forms were checked in the revised manuscript based on your comment and hope you find them satisfactory.
Comment 6: To improve the introduction and reference sections, you should follow quality papers.
[1] S. Cheng, S. Jianfu, M. Alrasheedi, P. Saeidi, A. R. Mishra, and P. Rani, "A new extended VIKOR approach using q-rung orthopair fuzzy sets for sustainable enterprise risk management assessment in manufacturing small and medium-sized enterprises," International Journal of Fuzzy Systems, vol. 23, no. 5, pp. 1347-1369, 2021.
[2] H. Li, S. Yin, and Y. Yang, "Some preference relations based on q‐rung orthopair fuzzy sets," International Journal of Intelligent Systems, vol. 34, no. 11, pp. 2920-2936, 2019.
[3] X. Peng, J. Dai, and H. Garg, "Exponential operation and aggregation operator for q‐rung orthopair fuzzy set and their decision‐making method with a new score function," International Journal of Intelligent Systems, vol. 33, no. 11, pp. 2255-2282, 2018.
Please go through those papers, and include and improve your literature review portion of the paper.
Response: Many thanks for this comment. This part was updated in the revised manuscript based on your comment and hope you find them satisfactory. Please see pages 6, lines 238-242.
The relevant text is copied below for you to review.
Cheng et al. presented a study for evaluating sustainable enterprise risk management in manufacturing small and medium-sized enterprises [31]. They adopted a new extended Vlse Kriterijumska Optimizacija Kompromisno Resenje (VIKOR) approach using q-ROFSs. Peng et al. introduced a study for presenting a new score function of q-ROFN for solving the failure issues when comparing two q-ROFNs [32].
Comment 7: Elaborate discussions of results. Try to point out each waveform using proper justification.
Response: Many thanks for this comment. Results were updated in the revised manuscript based on your comment and hope you find them satisfactory.
The relevant text is copied below for you to review.
Comment 8: Elaborate discussions of results. Try to point out each waveform using proper justification.
Response: Many thanks for this comment. Results were well discussed in the revised manuscript based on your comment and hope you find them satisfactory.
Comment 9: Rewrite the conclusion section in the summarized form.
Response: Many thanks for this comment. Results were updated in the revised manuscript based on your comment and hope you find them satisfactory.
The relevant text is copied below for you to review.
Given the spread of computer networks and the dependence of public and private institutions on their efficiency and quality of work, any disruption or sabotage of them may lead to great losses. Information systems and networks are constantly subject to cyber-attacks. Thus, firewalls and antivirus are not enough to fend off all these attacks, as they are only able to protect the "front entrance" of computer systems and networks. IDS can help protect your corporation from malicious activities. There are different types of IDS to protect networks, as intrusion attacks are becoming more and more common on a global scale. In addition, hackers using new technologies are trying to penetrate the system. IDS is a tool that identifies these attacks and will take an immediate step to get the system back to normal, as the IDS can also detect network traffic and send an alarm if a breach is found.
In this regard, this study discusses the most effective and used IDSs. This study was conducted with the participation of many experts under the q-ROF environment to deal with the uncertainty that may occur as a result of different circumstances and differences in evaluation frameworks. Six intrusion detection systems namely, Suricata ( ), Zeek ( ), Security onion ( ), Snort ( ), Wazuh ( ), and OSSEC ( ) have been evaluated according to four key criteria and thirteen sub-aspects. The main criteria namely, protected system, audit source location, targets, and types. The sub-aspects on the basis of which the effectiveness of intrusion detection systems was evaluated are: HIDS, NIDS, hybrids, host log files, network packets, application log files, IDS sensors alerts, applications, network, host, open-source, closed source, and freeware. A hybrid MCDM approach including q-ROF entropy-CoCoSo techniques was proposed, where entropy was applied to evaluate the main criteria and their sub-aspects. The CoCoSo method is applied to rate six IDSs according to their effectiveness. Afterward, comparative and sensitivity analyses were performed to confirm the stability, reliability, and performance of the proposed approach. The findings indicate that most of the IDSs appear to be systems with high potential. According to the results, Suricata is the best IDS that relies on multi-threading performance. Although the results here confirm that the proposed method is applicable and effective, it has some limitations. The key limitation of the approach is the difficult mathematical algorithm for the computation of Hamacher functions.
Comment 10: Some sentences should be rewritten.
Response: Many thanks for this comment. We have double-checked all the texts and rewritten the manuscript from the scratch.

Round 2
Reviewer 2 Report
The authors have replied to all the concerned comments. Therefore, paper can be accepted in its current form.
Author Response
Thank you very much for accepting our paper.
Reviewer 3 Report
Comment 4: What is the computation time for the algorithm? Provide running time for the proposed method? Provide the comparison of computation time between the proposed method and other works.
How were results achieved without making a prototype ? i need running time , not complexity or memory analysis
Comment 5: If the proposed method is highly accurate, can you please provide some failure cases for the method? And discuss the underlying cause of these failure cases.
These two comments are not answered by the authors
Author Response
Comment 4: What is the computation time for the algorithm? Provide running time for the proposed method? Provide the comparison of computation time between the proposed method and other works.
How were results achieved without making a prototype? i need running time, not complexity or memory analysis
Reply: the topic of the paper is related to MCDM NOT metaheuristic algorithms, the core of the paper proposed a model for Appraising Intrusion-Detection System for Network Security Communications under uncertainty that may occur as a result of different circumstances and differences in evaluation frameworks. Six intrusion detection systems namely, Suricata (IDS_1), Zeek (IDS_2), Security onion (IDS_3), Snort (IDS_4), Wazuh (IDS_5), and OSSEC (IDS_6) have been evaluated according to four key criteria and thirteen sub-aspects
Comment 5: If the proposed method is highly accurate, can you please provide some failure cases for the method? And discuss the underlying cause of these failure cases.
Reply: as same as the above reply